# Causes of growing middle-upper tropospheric ozone over the Northwest Pacific region

Xiaodan Ma[1,2], Jianping Huang[3,4], Michaela I. Hegglin[2,5], Patrick Jöckel[6], and Tianliang Zhao[1]

[1]Collaborative Innovation Center on Forecast and Evaluation of Meteorological Disasters, Key Laboratory for Aerosol-Cloud-Precipitation of China Meteorological Administration, Nanjing University of Information Science and Technology, Nanjing 210044, China.
[2]Institute of Energy and Climate Research – Stratosphere (IEK-7), Forschungszentrum Jülich, Jülich, Germany.
[3]Lynker, Environmental Modeling Center, NOAA National Centers for Environmental Prediction, College Park, MD, USA.
[4]Center for Spatial Information Science and Systems, College of Science, George Mason University, Fairfax, VA 22030, USA.
[5]Department of Meteorology, University of Reading, Reading, United Kingdom
[6]Deutsches Zentrum für Luft- und Raumfahrt (DLR), Institut für Physik der Atmosphäre, Oberpfaffenhofen, Germany

*Correspondence to*: Jianping Huang (jianping.huang@noaa.gov)

**Abstract.** Long-term ozone ($O_3$) changes in the middle to upper troposphere are critical to climate radiative forcing and tropospheric $O_3$ pollution. Yet, these changes remain poorly quantified through observations in East Asia. Concerns also persist regarding the data quality of the ozonesondes available at the World Ozone and Ultraviolet Data Center (WOUDC) for this region. This study aims to address these gaps by analyzing $O_3$ soundings at four sites along the northwestern Pacific coastal region over the past three decades, and assessing their consistency with an atmospheric chemistry-climate model simulation. Utilizing the European Centre for Medium-Range Weather Forecasts (ECMWF) – Hamburg (ECHAM)/Modular Earth Submodel System (MESSy) Atmospheric Chemistry (EMAC) nudged simulations, it is demonstrated that trends between model and ozonesonde measurements are overall consistent, thereby gaining confidence in the model's ability to simulate ozone trends and confirming the utility of potentially imperfect observational data. A notable increase in $O_3$ mixing ratio around 0.29-0.82 ppb $a^{-1}$ extending from the middle to upper troposphere is observed in both observations and model simulations between 1990 and 2020, primarily during spring and summer. The timing of these $O_3$ hotspots is delayed when moving from south to north along the measurement sites, transitioning from late spring to summer. Investigation into the drivers of these trends using tagged model tracers reveals that ozone of stratospheric origin ($O_3S$) dominates the absolute $O_3$ mixing ratios over the middle-to-upper troposphere in the subtropics, contributing to the observed $O_3$ increases by up to 96% (40%) during winter (summer), whereas ozone of tropospheric origin ($O_3T$) governs the absolute value throughout the tropical troposphere and contributes generally much more than 60% to the positive $O_3$ changes, especially during summer and autumn. During winter and spring, a decrease of $O_3S$ is partly counterbalanced by an increase of $O_3T$ in the tropical troposphere. This study highlights that the enhanced downward transport of stratospheric $O_3$ into the troposphere in the subtropics and a surge of tropospheric source $O_3$ in the tropics are the two key factors driving the enhancement of $O_3$ in the middle-upper troposphere along the Northwest Pacific region.

**Keywords:** EMAC model, ozone sounding, stratospheric intrusion, tropospheric ozone

## 1. Introduction

Stratospheric intrusions and photochemical production are two major contributors to tropospheric ozone ($O_3$, Ding and Wang, 2006; Neu et al., 2014; Williams et al., 2019; Zhao et al., 2021). The stratosphere accommodates 90% of the total $O_3$ in the atmosphere. As the largest natural source, downward transport of $O_3$-enriched air from the stratosphere exerts an important impact particularly on the seasonality of tropospheric $O_3$ (Williams et al., 2019). Tropospheric $O_3$ increases of 7% (measured as a partial column between 3-9 km) between 2005 and 2010 over China have been identified as a consequence of increased $O_3$ precursor emissions and enhanced downward transport from stratospheric $O_3$ (Verstraeten et al., 2015). While photochemical production is highly dependent on anthropogenic emissions, the impact of stratospheric intrusions on tropospheric $O_3$ is mainly governed by inter-annual variability and climate-driven changes in the atmospheric circulation (Neu et al., 2014; Albers et al., 2018). Compared to the spatiotemporal variations of $O_3$ in the lower troposphere, the counterpart in the middle-upper troposphere and their underlying causes remain inadequately quantified, largely due to scarcity of long-term, vertically resolved observational data.

Chemistry-climate modeling studies demonstrate that climate variability in the atmospheric circulation such as Brewer-Dobson circulation promotes stratospheric intrusions and enhances $O_3$ abundance in the upper troposphere (Sudo et al., 2003; Young et al., 2018; Akritidis et al., 2019; Griffiths et al., 2020; Liao et al., 2021). A study with a stratospheric chemistry-climate model projects a 20−30% increase in global stratosphere-to-troposphere transport (STT) $O_3$ flux from 1965 to 2095, as the result of an accelerated stratospheric Brewer-Dobson circulation under an intermediate climate change scenario (Hegglin and Shepherd, 2009). Furthermore, chemistry-climate models (CCMs) predict an even larger increase of the STT $O_3$ flux (25−80%) under climate change scenarios such as RCP8.5 (Collins, 2003; Sudo et al., 2003; Meul et al., 2018). Notably, Williams et al. (2019) identified an enhanced STT $O_3$ over Asia and the Pacific region during 1980-2010 based on two different CCMs. Several small-scale processes in proximity to the tropopause lead to irreversible STT events, including Rossby wave breaking, tropospheric cyclones, cut-off lows, and tropopause folding events (Holton et al., 1995). On a regional basis, including East Asia and its coastal area, subtropical westerly jets modulate the location, timing, and frequency of tropopause folds (Sprenger et al., 2003; Albers et al., 2018). Satellite measurements of $O_3$ and water vapor over six years were used to quantify the impact of a changing stratospheric circulation on tropospheric $O_3$ in the northern hemisphere (Neu et al., 2014). These observation-based results support the modeling studies that the intensified stratospheric Brewer-Dobson circulation tends to enhance the impact of the stratospheric intrusions on tropospheric $O_3$. However, the conclusions drawn from the numerical studies have not yet been validated through long-term $O_3$ measurements, particularly $O_3$-sounding data (Trickl et al., 2011).

From 1990 onwards, a significant amount of the anthropogenic emissions responsible for $O_3$ formation have shifted from North America and Europe to Asia (Granier et al., 2011; Cooper et al., 2014; Zhang et al., 2016). In East Asia, the overall long-term trend of the daytime average near-surface $O_3$ is 0.45 ppb $a^{-1}$, contrasting with a trend of − 0.28 ppb $a^{-1}$ in North America in the summertime (April-September) during 2000-2014 (Chang et al., 2017). Several studies have documented the increase in emissions of $O_3$ precursors at few sites available for evaluating the long-term trends across East Asia (Ma et al., 2016; Sun et al., 2016; Xu et al., 2016; Wang et al., 2017). On the other hand, some regions in East Asia have seen a decline in precursor emissions after 2004, such as Beijing, Hong Kong, and Japan due to local emission control efforts (Krotkov et al., 2016; Liu et al., 2016; Miyazaki et al.,

2017; van der A et al., 2017). Elevated $NO_2$ emissions over megacities in China were possibly transported to Japan, potentially offsetting the local emission control efforts (Duncan et al., 2016). Further research is required to understand the long-term changes in tropospheric $O_3$, especially in East Asia, where rapid economic growth coincides with strict environmental regulations.

In this study, we present thirty years of $O_3$ observations from balloon soundings at fine vertical resolution (less than 10m) with a focus on latitudinal differences. To this end, observations from four sounding sites are analyzed together with model simulation results to quantify the long-term trends of middle-upper tropospheric $O_3$ and contributions of different origins along the northwestern Pacific coastal region. We are particularly interested in the regional difference near 30°N, the transition zone between the Hadley and Ferrel circulation cells, where the subtropical jet (STJ) prevails and tropopause folding is frequently observed (Škerlak et al., 2015; Zhao et al., 2021). The specific questions to be addressed by this study are 1) How do $O_3$ trends in the middle-upper troposphere vary with latitude and season over the northwestern Pacific coastal regions and are these observed trends consistent with those derived from a chemistry-climate model? 2) To what extent are these tropospheric $O_3$ changes linked to stratospheric influences? And 3) to what extent are these tropospheric $O_3$ changes linked to tropospheric sources, i.e. photochemical ozone production due to biogenic and anthropogenic precursor emissions? The study aims to provide observational evidence to validate and constrain the CCMs' predictions of climate-change impact on tropospheric $O_3$ in East Asia (e.g., Williams et al., 2019) where such information is still lacking.

## 2. Data and method

### 2.1 Ozonesonde observations

Around thirty years of $O_3$-sounding data at four sites along the northwestern Pacific coastal regions (Sapporo, Tsukuba, Naha, and Hong Kong) are used to characterize spatiotemporal variations of $O_3$ in the troposphere. Ozonesondes were launched around 14:00 local standard time (LST) once a week, which corresponds to the time when photochemical production reaches its daily maximum (Oltmans et al., 2004). The ozonesonde measurements include $O_3$ partial pressure, temperature, relative humidity, wind speed, and wind direction. Vertical $O_3$ measurements range from the surface to the middle stratosphere approaching 30 km. The Hong Kong site has continually operated the electrochemical concentration cell (ECC) instrument since the beginning of its record. For the three sites in Japan, the $O_3$-sounding data were measured by Carbon-iodine (CI) ozonesondes with 10-second recording intervals before 2009 and changed to the ECC instrument with 2-second recording intervals. The application of correction factors on ozone profiles during the CI measurement period has been found to inaccurately influence the tropospheric $O_3$ (Morris et al., 2013). We removed the applied correction factor on the original ozonesonde data from WOUDC at three Japanese-sounding stations hereinafter. The operating principle of CI ozonesondes and ECC ozonesondes both are based on the reaction of $O_3$ to potassium iodide solution wherein free iodine is liberated (Johnson et al., 2002; Witte et al., 2018). However, the transition of the measurement technology from CI to ECC around 2009 could lead to an overestimation of uncertainties on the long-term $O_3$ trends. The research from the cross-evaluation of OMI data and the ozonesonde observation in Japan sites shows that CI ozonesonde measurements are negatively biased relative to ECC measurements by 2–4 DU compared with the OMI data (Bak et al., 2019). Removing the correction factor in the CI measurements can improve the consistency of ozonesondes with OMI data (Morris et al. 2013). It is worth noting that the conclusion we draw from current available long-term ozonesonde observation has limitations on the long-term trends but still has

important implications on the understanding of tropospheric $O_3$ changes and model evaluations. The weekly launch frequency of the ozonesondes has been validated as reliable in representing long-term $O_3$ trends, as evidenced by comparing them with near-surface $O_3$ trends at hourly time resolution (Liao et al., 2021). A summary of ozonesonde-site location and data availability is presented in Table 1 and Figure 1.

We limit our analyses of tropospheric and lower-stratospheric $O_3$ profiles to altitudes below 18 km and remove duplicate $O_3$ values at the same heights in the time series to prevent redundant measurements during the up-and-downs of the floating $O_3$ sounding balloons. $O_3$ profiles with continuous data missing more than a 200m vertical coverage are excluded. The selected valid $O_3$ profiles with 10s or 2s recording intervals are linearly interpolated into 10m vertical intervals and then averaged into 50m data points. The $O_3$ profiles after the quality control with 50m vertical resolution are used for further analysis.

Due to the latitudinal differences and the seasonal variations in tropopause height across the four $O_3$-sounding observation sites, it is inappropriate to apply a specific height as the tropopause height. We thus employ the World Meteorological Organization lapse rate tropopause definition to calculate the tropopause height (hereafter called $Z_t$) for each site and $O_3$ profile. The $Z_t$ is defined as the level at which the lapse rate decreases to 2 K km$^{-1}$ or less, provided that the average lapse rate between this level and all higher levels within 2 km does not exceed 2 K km$^{-1}$ (WMO, 1957).

To better compare $O_3$ levels and trends at different latitudes within the troposphere, we normalize the height of each $O_3$ profile into 0~1 by dividing the altitude by the tropopause height $Z_t$. The upper troposphere (UT) is then defined by the normalized height ($Z/Z_t$) range between 0.7 and 0.9. The middle troposphere (MT) and lower troposphere (LT) are 0.4~0.6 and 0~0.2 $Z/Z_t$, respectively.

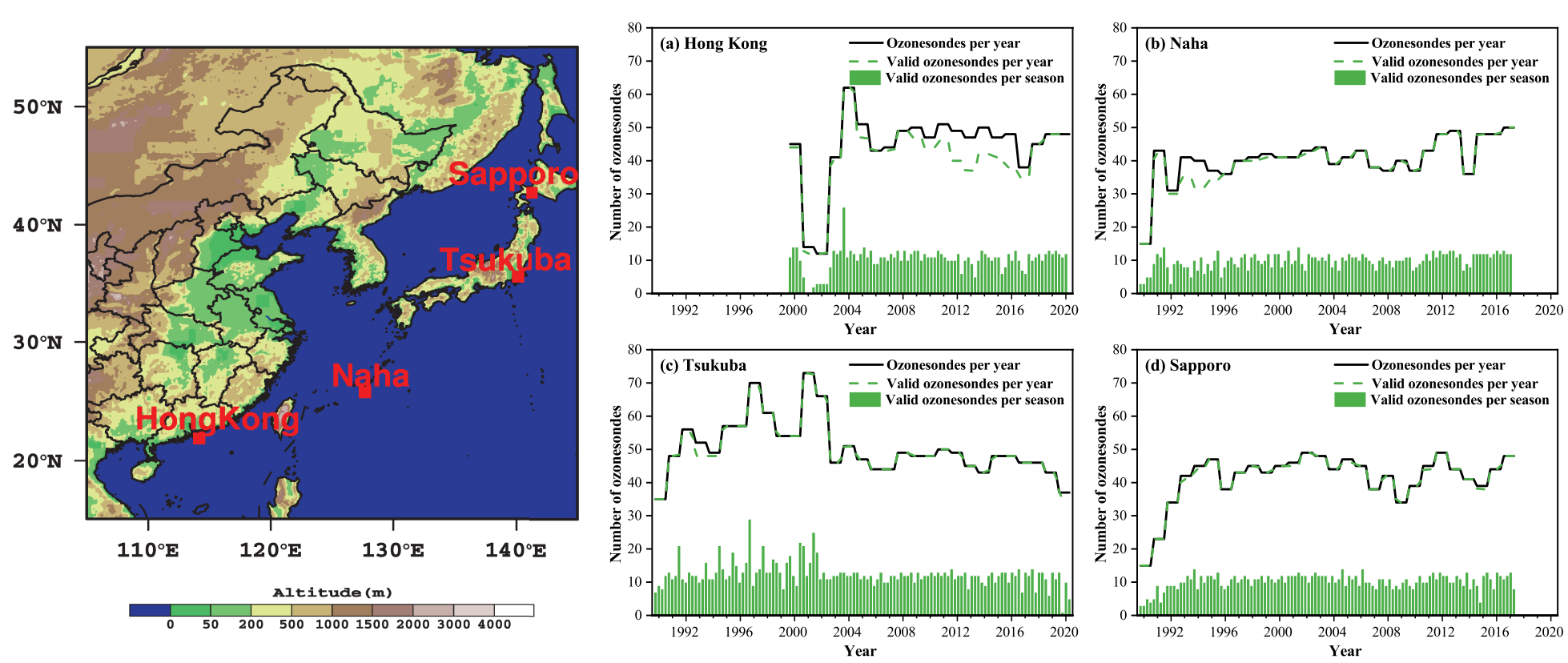

**Figure 1. Location of $O_3$-sounding sites and seasonal and annual ozonesonde sampling at a) Hong Kong, (b) Naha, (c) Tsukuba, and (d) Sapporo. The continuous line shows the number of ozonesondes launched per year. The bars show**

**the corresponding number per season. The dashed line indicates the number of valid ozonesondes reaching up to 18 km altitude.**

**Table 1. Location of $O_3$-sounding sites, measurement periods, and total data available along the northwestern Pacific coastal region.**

| Station | Latitude | Longitude | Elevation (m) | Period | Total data | Valid data (18km) |
|---|---|---|---|---|---|---|
| Sapporo | 43.10°N | 141.30°E | 19 | 1990-2017 | 1167 | 1159(99%) |
| Tsukuba | 36.06°N | 140.13°E | 31 | 1990-2020 | 1564 | 1556(99%) |
| Naha | 26.20°N | 127.70°E | 27 | 1990-2017 | 1137 | 1114(98%) |
| Hong Kong | 22.31°N | 114.17°E | 66 | 2000-2020 | 929 | 863(93%) |

## 2.2 EMAC model and simulation setup

In this study, the European Centre for Medium-Range Weather Forecasts (ECMWF) – Hamburg (ECHAM)/Modular Earth Submodel System (MESSy) Atmospheric Chemistry (EMAC) model is utilized to investigate the long-term changes of tropospheric $O_3$ and to quantify the relative contributions of different driving factors. The EMAC model is a global model that considers the interaction of chemistry and dynamic processes between the surface and the middle atmosphere (Jöckel et al., 2016). The REF-D1-specific dynamics (SD) simulation results from the EMAC model are used in this study. The REF-D1 experiment is a hindcast simulation of the atmospheric state, using a prescribed sea surface temperature and sea ice from observations along with forcing for the extra-terrestrial solar flux, long-lived greenhouse gasses, and $O_3$-depleting substances, stratospheric aerosols, and an imposed quasi-biennial oscillation that approximate the observed variations over the historical period to the fullest extent possible (Jöckel, 2023). The hindcast simulations are performed from 1980 to 2019 with the SD nudging by Newtonian relaxation towards ECMWF ERA-5 reanalysis meteorological data (Hersbach et al., 2020), including temperature, logarithm of surface pressure, divergence, and vorticity.

The simulations are conducted at a T42 (triangular) spectral resolution corresponding to an approximately 2.8° × 2.8° quadratic Gaussian grid, 90 hybrid sigma pressure vertical levels from surface up to 0.01 hPa, and with a 720s time step length (Jöckel et al., 2016). EMAC uses chemical submodels, the Module Efficiently Calculating the Chemistry of Atmosphere (MECCA, Sander et al., 2011) and the scavenging submodel (SCAV, Tost et al., 2006) to describe comprehensive chemical reaction mechanisms in gas and liquid phases that include $O_3$, $CH_4$, $HO_x$ and $NO_x$ chemistry, non-methane hydrocarbon (NMHC) chemistry up to $C_4$ and isoprene, halogen (Cl and Br) chemistry, and sulfur chemistry.

Emissions of lightning NOx, soil NOx, and isoprene (C5H8) are calculated online for EMAC using the submodels LNOx (Tost et al., 2007) and ONEMIS (Kerkweg et al., 2006; Jöckel et al., 2016), respectively. EMAC simulates the photolysis (submodel JVAL, Sander et al., 2014) and shortwave radiation schemes (FUBRAD, Kunze et al., 2014) consistently, with particular regard to the evolution of the 11-year solar cycle (Morgenstern et al., 2017). For anthropogenic emissions, mixing ratios of greenhouse gases, ozone-depleting substances (ODS), and other boundary conditions, the EMAC model setup follows the CCMI-2020 protocol of the refD1 hindcast simulations (SPARC, 2021).

The EMAC model provides the diagnostic tracer $O_3S$ to directly measure the stratosphere-to-troposphere exchange of $O_3$. The $O_3S$ tracer is transported across the tropopause into the troposphere and is removed by tropospheric $O_3$

reactions (Jöckel et al., 2006; Jöckel et al., 2016). When $O_3S$ re-enters the stratosphere, it is re-initialized (Roelofs and Lelieveld, 1997). The tropospheric $O_3$ source ($O_3T$) is here calculated as tropospheric $O_3$ minus stratospheric $O_3$ ($O_3T = O_3 - O_3S$).

To better compare the model results with the observations, the simulation data is extracted from the grid boxes nearest to the observation sites. Specifically, 200 hPa is chosen for Hong Kong and Naha, and 400 hPa for Tsukuba and Sapporo to represent the upper troposphere. The middle troposphere is defined at 500hPa, while the lower troposphere is represented by 850 hPa in the model results. To assess the statistical significance of the differences, a paired two-sided t-test ($p<0.05$) is conducted for comparison.

## 3. Results

### 3.1 Observational changes at different stations

#### 3.1.1 Climatological distribution of tropospheric ozone

Figure 2 depicts the climatologically vertically resolved tropospheric $O_3$ distribution with respect to months. The four sites all show a distinct tongue-shaped pattern in top-down direction characterized by high concentrations of $O_3$ greater than 70 ppb, each exhibiting peak levels in distinct months. The ozone tongue extends from the lower stratosphere to the middle troposphere, even further spreading downward to the lower troposphere. In subtropical regions such as Hong Kong and Naha, the ozone tongue starts to appear in early spring. Their appearance becomes progressively delayed when moving towards higher latitudes, with peak occurrences observed in Tsukuba during June and Sapporo in July (Figure 2c-d). For the mid-latitudes over the Pacific region, the incidence of stratospheric intrusions has been found to have a strong correlation with the location of the STJ (Zhao et al., 2021). The northward shift of the STJ with seasons agrees well with the occurrence of ozone tongue in different months over the four sites along the northwest Pacific coastal regions (Figure S2). The tropopause folding on the south part of the STJ could lead to more stratospheric intrusion contributions to the ozone tongue. This suggests a potential contribution of stratospheric intrusion to the seasonal lag of the ozone tongue.

On the other hand, the four sites display distinct month-height cross-section distribution patterns of $O_3$. In near-tropical regions such as Hong Kong and Naha during the summer, a relatively "clean" layer with $O_3$ mixing ratios less than 40 ppbv extends from the surface to about 5.0 km above the ground level (AGL). Such a structure, characterized by low concentrations in the lower troposphere is not observed at the other two high-latitude sites. The unfavorable meteorological conditions linked to the East Asian monsoon like a strong wind, precipitation, and less radiation could lead to significant ozone scavenging and less photochemical production. This suggests that the East Asian summer monsoon has a more significant impact on $O_3$ vertical structures at lower latitude sites compared to high latitude sites. Meanwhile, it is noticed that high $O_3$ mixing ratios appear within the atmospheric boundary layer (ABL) (0.7-1.6km according to Su et al., (2017)) in Hong Kong in autumn (Figure 2a), which represents the combined effect of local emissions and regional transport. During this season, the prevailing winds are predominantly from northwest to north, which could bring elevated levels of $O_3$ and its precursors from the Pearl River Delta region, a major manufacturing base in China, to Hong Kong (Ding et al., 2013; Lin et al., 2021).

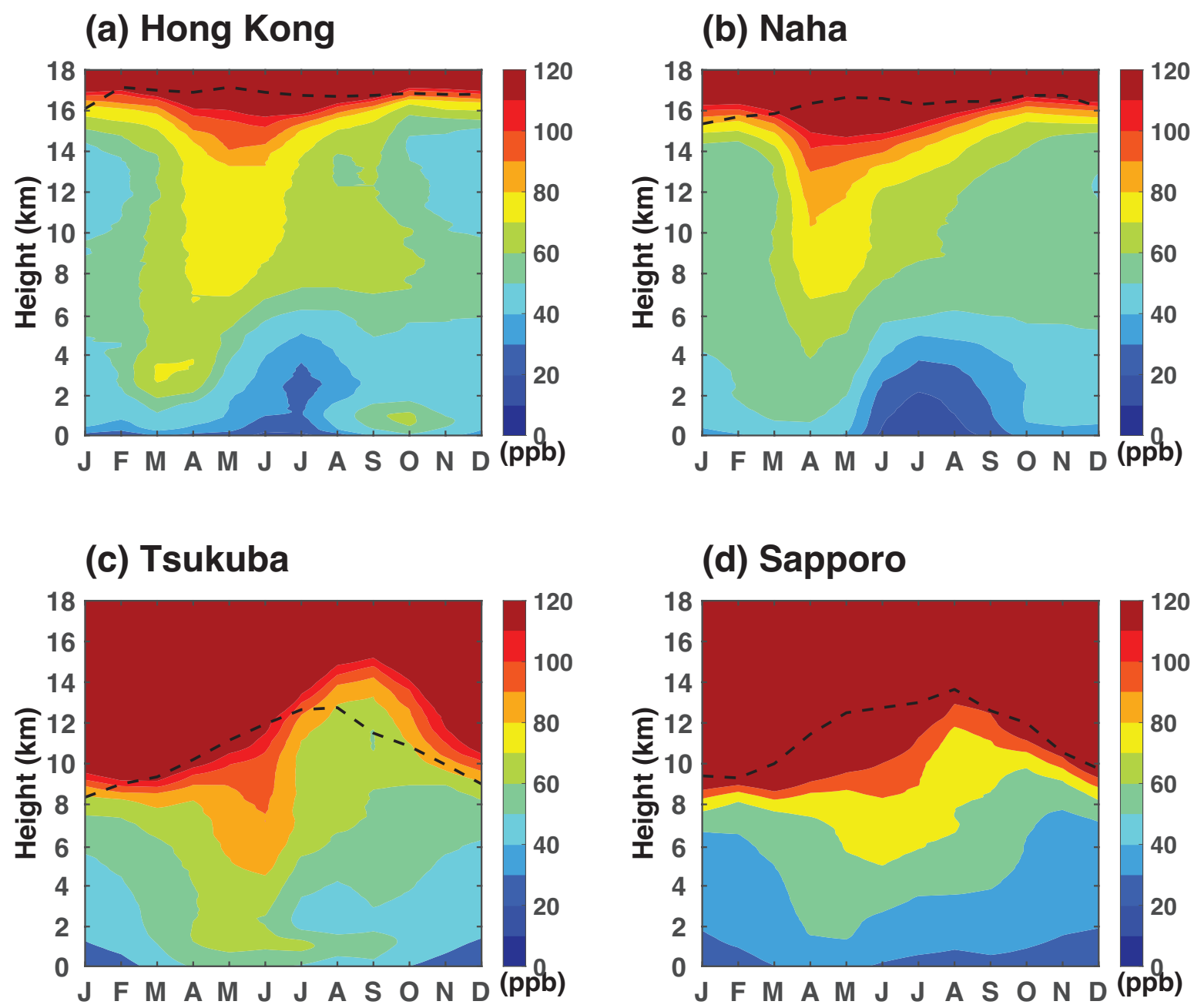

**Figure 2. Month-height cross sections of monthly mean $O_3$ at four $O_3$-sounding sites, (a) Hong Kong, (b) Naha, (c) Tsukuba, and (d) Sapporo, from 1990 to 2020 (2000 to 2020 for Hong Kong). Black dash lines indicate the multi-year average tropopause height.**

### 3.1.2 Long-term trends in different layers of the troposphere

Figure 3 presents the long-term trends of $O_3$ in the upper, middle, and lower troposphere. In general, $O_3$ in the upper troposphere shows larger increases during boreal spring and summer than autumn and winter among the four sites except for Hong Kong. The largest $O_3$ trends are observed at Naha with an increase of 0.82 ppb $a^{-1}$ during the summer and at Tsukuba (0.63 ppb $a^{-1}$) during the spring (at a 95% confidence level). Hong Kong only shows a significant $O_3$ increase in spring with 0.60 ppb $a^{-1}$ while Tsukuba exhibits extensive $O_3$ increase except winter. For the Sapporo site, substantial positive $O_3$ changes are observed during summer but not statistically significant due to large temporal variabilities. This finding implies the importance of STJ in the change of $O_3$ in the upper troposphere at Naha and Tsukuba. The locations are situated within the transitional zone between the Hadley and Ferrel circulation cells in spring and summer, as illustrated in Figure S2. This influence appears more pronounced in comparison to the other two sites, namely Hong Kong and Sapporo, which are situated further from this transitional zone.

Moving to the middle troposphere, Naha and Tsukuba consistently display an ozone increase during all four seasons. The changes at these two sites in spring, summer, and autumn are more evident than those at the other two sites and winter. This suggests a potential strengthened contribution from regional transport and stratospheric intrusion for these two sites. In addition, lightning-produced $NO_x$ emissions contribute to major events of $O_3$ in the middle-upper troposphere over convection active regions (Liu et al., 2002; Zhang et al, 2012). How those factors contribute to $O_3$ enhancement remains a question for further investigations.

In the lower troposphere, substantial $O_3$ increases are observed at all sites in spring except Tsukuba. $O_3$ enhancement in the lower troposphere over Hong Kong during springtime is associated with either equatorial Northern Hemisphere biomass burning in Africa or Southeast Asian biomass burning (Oltmans et al., 2004). The Tsukuba site experienced a slight decrease in summer over the past three decades. Such a decrease could be primarily attributed to the changes in anthropogenic emissions in East Asia (Li et al, 2019).

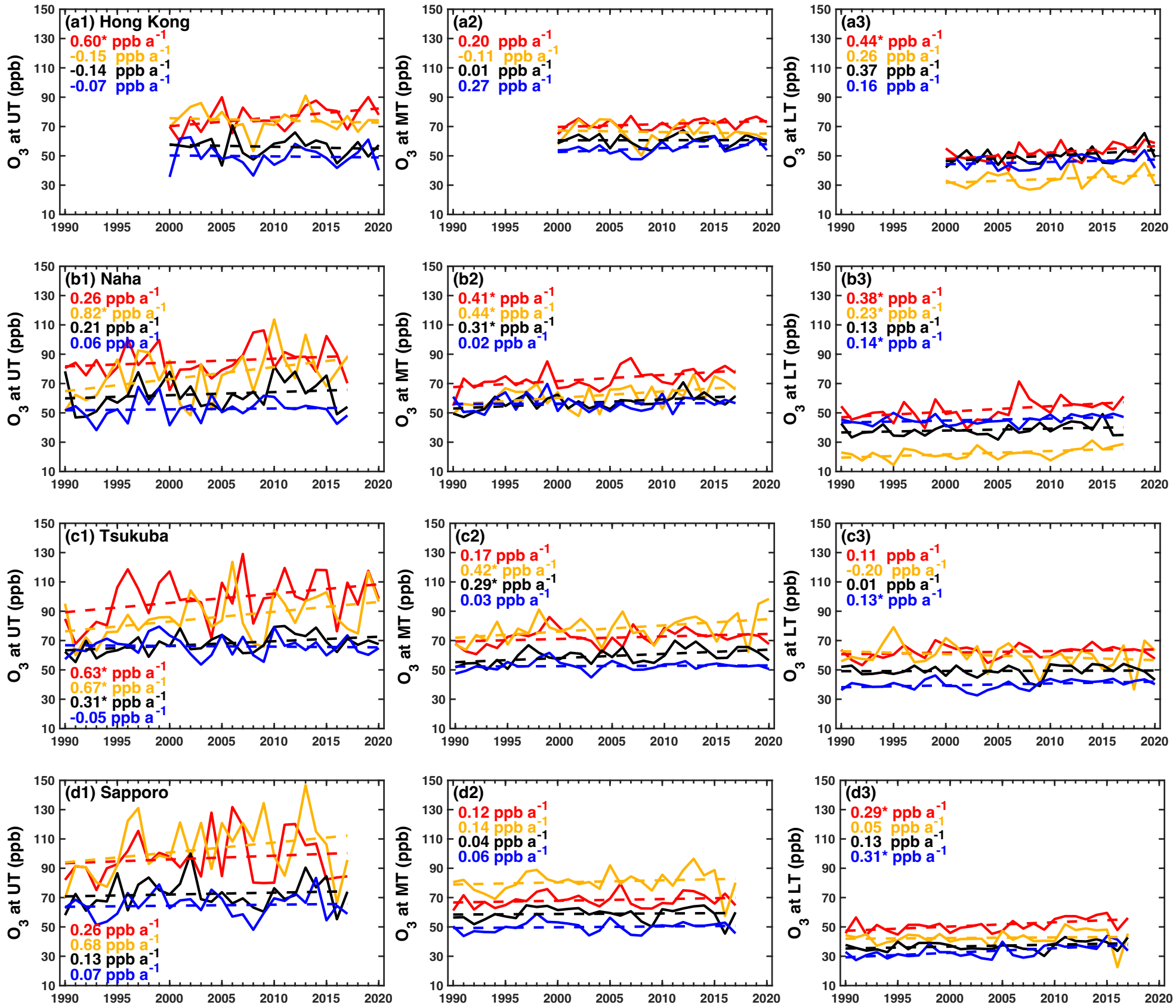

**Figure 3. Long-term changes of $O_3$ in the Upper Troposphere (first column), Middle Troposphere (second column), and Lower Troposphere (third column) in boreal spring (MAM, red lines), summer (JJA, yellow lines), autumn (SON, black lines), and winter (DJF, blue lines) at Hong Kong (a1-a3), Naha (b1-b3), Tsukuba (c1-c3), and Sapporo (d1-d3). Trends with a star symbol (*) indicate significance at the 95% confidence level.**

Overall, the long-term changes in tropospheric $O_3$ displayed considerable variability, contingent on the atmospheric layers (i.e., low, middle, and upper) and the geographical latitude of observation sites. Naha, Tsukuba, and Sapporo exhibited an increase in the middle-upper troposphere. A substantial rise is observed in the upper troposphere during summer over Naha (0.82 ppb $a^{-1}$) and spring over Tsukuba (0.63 ppb $a^{-1}$). When compared to the other three sites, changes in the middle-upper troposphere over Hong Kong are smaller or negative, except during springtime. All four sites demonstrated an increase in $O_3$ mixing ratios across the four seasons in the lower troposphere, except for summer in Tsukuba. Investigating the driving factors behind such differences in change becomes one of the objectives of this study. A more comprehensive exploration of $O_3$ origin and their contributions

to the changes in tropospheric $O_3$ will be discussed in Section 3.2, leveraging modeling results to provide deeper insight.

**3.1.3 Changes in composite $O_3$ cross-sections between decades**

Tropospheric $O_3$ shows a larger variability in the upper troposphere compared to the middle and lower troposphere (Figure 3 a1-d3). Such a large variability, likely driven by transport and dynamics in the tropopause region, impedes drawing definite conclusions on long-term trends for single measurement sites with infrequent sampling. Therefore, the aggregation of tropospheric $O_3$ during the early and late decades is expected to provide more robust insights.

Figure 4 illustrates the vertically resolved tropospheric $O_3$ distributions and changes between the early (the 1990s for Naha, Tsukuba, and Sapporo; the 2000s for Hong Kong) and late (2010s) decades as a function of the month. Their respective tropospheric $O_3$ changes over the same period (i.e., 2000s to 2010s) at the four sites are presented in Figure S1 to demonstrate the consistency of the results. The time lag pattern for the ozone tongue remains the same from April in the southern site of Hong Kong to July in the northern site of Sapporo for the first and the last decades (Figure 4 a1-d1). However, there are noticeable increases in $O_3$ mixing ratios and a deeper layer extension of the $O_3$ concentration greater than 80 ppbv from the stratosphere to the troposphere at Naha and Tsukuba over the past several decades (Figure 4 a2-d2).

As illustrated in Figure 4 a3-d3, Naha, Tsukuba, and Sapporo exhibit significant enhancements of $O_3$ from the middle-upper troposphere to the lowermost stratosphere, ranging from 20 to 40 ppb. In contrast to the three sites in Japan, Hong Kong shows more significant $O_3$ changes in the lower troposphere. The build-up of lowermost stratospheric (LMS) $O_3$ happens from the winter to spring, thus the STE flux of $O_3$ normally reaches its peak during late spring to early summer in the extratropical regions (e.g., Škerlak et al., 2015; Albers et al., 2018). The ozone tongue during the spring and summer is possibly associated with enhanced contribution from stratospheric intrusions. While it may be tempting to conclude that such an $O_3$ increase primarily originates from the stratosphere due to their proximity, observational data alone cannot provide a definite conclusion. Additionally, different locations among the four sites may introduce further differences in $O_3$ sources.

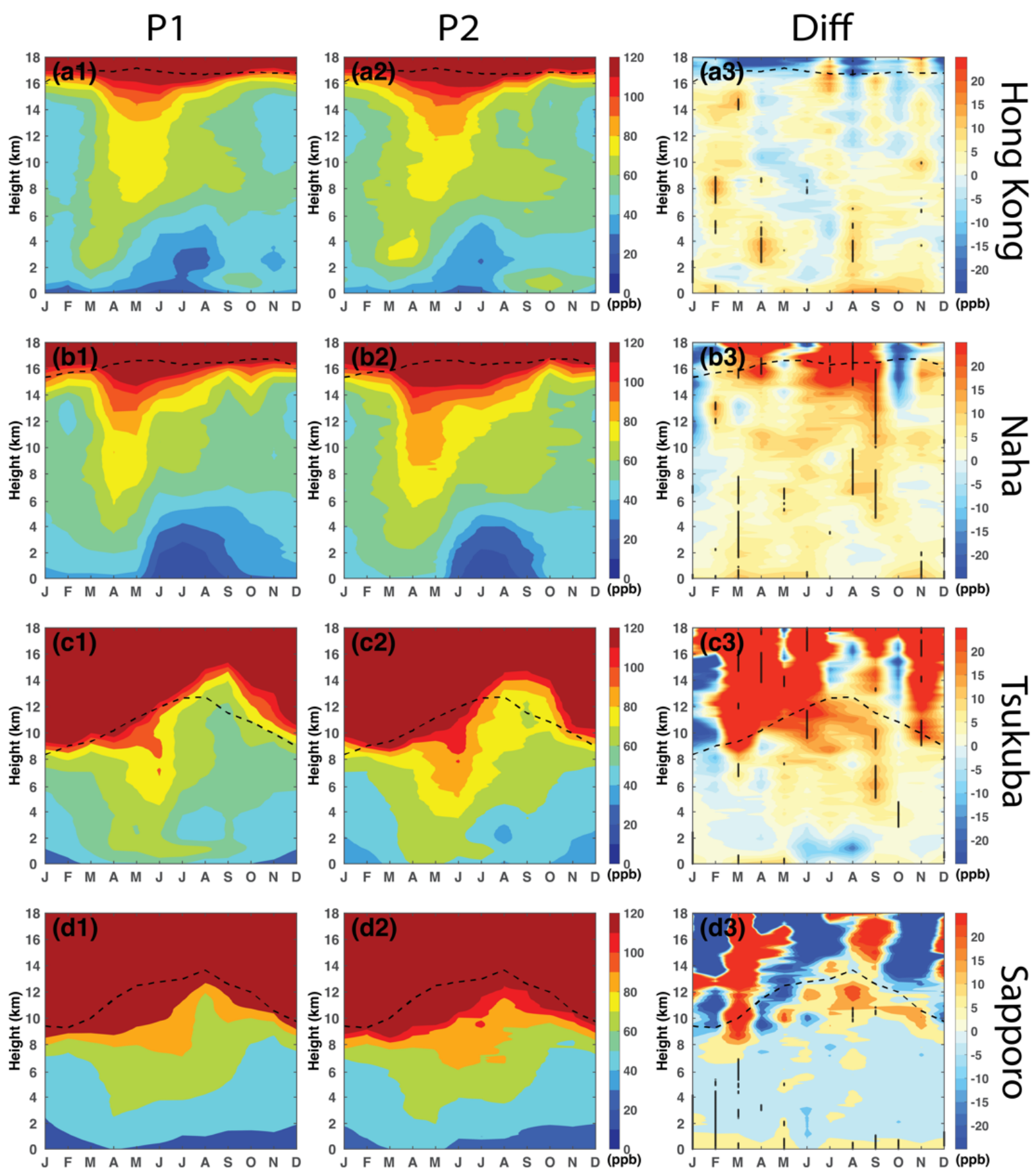

**Figure 4. Month-height cross sections of monthly mean composite $O_3$ in the first period P1 (1990s for Naha, Tsukuba, and Sapporo, but 2000s for Hong Kong), the last period (P2: 2010s), and the differences of $O_3$ between P2 and P1 at (a1-a3) Hong Kong, (b1-b3) Naha, (c1-c3) Tsukuba and (d1-d3) Sapporo. Black dash lines indicate the tropopause heights. Dashed lines in the i-l represent the layer with statistically significant changes according to a paired two-sided t-test ($p < 0.05$).**

Figures 5b-d present a comparison of seasonally-averaged vertical $O_3$ profiles between the 1990s and the 2010s at the Naha, Tsukuba, and Sapporo sites. A parallel analysis is conducted for Hong Kong but for a comparison between the 2000s and 2010s (Figure 5a). While the general trend indicates an increase of $O_3$ mixing ratios with altitude, with higher values during spring and summer, several noteworthy features are identified from Figure 5. Firstly, vertical $O_3$ profiles vary with latitude and season. For instance, Hong Kong and Tsukuba show $O_3$ peaks within the ABL in autumn (black lines) and during summer (yellow lines), respectively. These peaks suggest a predominant influence of local anthropogenic emissions during the warmer months. A substantial $O_3$ peak at Hong Kong is observed around 0.2 normalized height (around 3-4 km above ground level) during spring. This enhancement is attributed to a combination of stratospheric intrusions and the transboundary transport of biomass-burning emissions originating from Southeast Asia (Liao et al., 2021; Zhao et al., 2021). On the other hand, Naha

and Sapporo do not exhibit discernible peaks in the lower troposphere, suggesting a relatively smaller impact from the combination of near-surface factors and stratospheric intrusions.

Secondly, the seasonal minimum $O_3$ mixing ratios in the lower troposphere are observed during summer rather than winter, contrasting with the middle to upper troposphere observations over Hong Kong and Naha. This seasonal difference in the lower troposphere could be attributed to the influence of the East Asia Monsoon as discussed earlier, while not so clear for the seasonal difference in the middle-upper troposphere. Conversely, the minimum seasonal $O_3$ mixing ratios occur during winter throughout the entire troposphere at the other two sites.

Thirdly, enhancements of $O_3$ in the middle and upper troposphere are considerably more pronounced over Naha, Tsukuba, and Sapporo than over Hong Kong during the warm seasons (spring and summer) over the past three decades. This enhancement is particularly significant in the upper troposphere in Naha and Tsukuba during summer, as indicated by the dashed and solid yellow lines. In Hong Kong, enhancements are primarily observed at the top of the ABL in spring and within the ABL in fall, corresponding to where seasonal maxima are observed. These findings align with previous research (Huang et al., 2005; Ding et al., 2013; Liao et al., 2021; Lin et al., 2021).

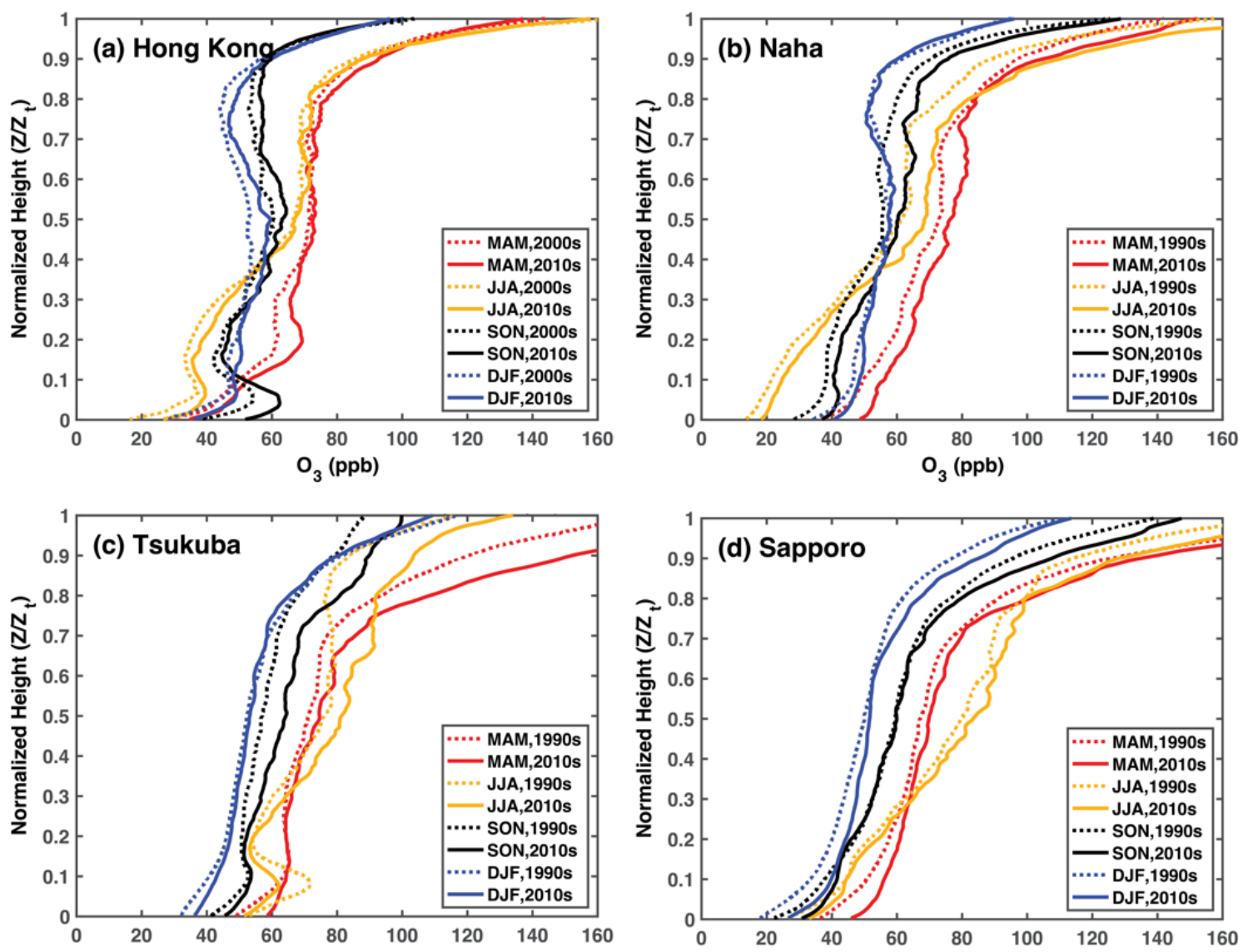

**Figure 5. A comparison of vertical profiles of seasonal mean $O_3$ during spring (red), summer (yellow), autumn (black), and winter (blue) at four sites (a) Hong Kong, (b) Naha, (c) Tsukuba, and (d) Sapporo between the first and the latest decades.**

### 3.2 Quantification of stratospheric intrusion versus tropospheric production using EMAC simulations

In order to substantiate the observational findings, we now turn to the quantification of the relative contributions of key drivers to the observed changes in tropospheric $O_3$ based on the EMAC simulations.

#### 3.2.1 Evaluation of EMAC simulations

The EMAC simulations of $O_3$ at different portions of the troposphere are further evaluated with the $O_3$ sounding data during the study period. As illustrated in Figure 6, the majority of data points are located above the 1:1 line at all sites, indicating that the EMAC over-predicts $O_3$ in the troposphere, which agrees with other related studies (Jöckel et al., 2016; Young et al., 2018; Revell et al 2018). Meanwhile, the EMAC model shows a better representation in the upper and lower troposphere than the middle troposphere in Hong Kong and Naha, as indicated by the coefficient of determinations ($R^2$). For instance, $R^2$ reaches to the highest value of 0.75 in the lower troposphere over Naha (Figure 7c2), whereas $R^2$ is only about 0.23 for the middle troposphere over Hong Kong (Figure 7 b1). As for the mid-latitude sites, Tsukuba and Sapporo, the EMAC model shows a relatively good representation of $O_3$ in the different layers of the troposphere, despite the overall overestimation.

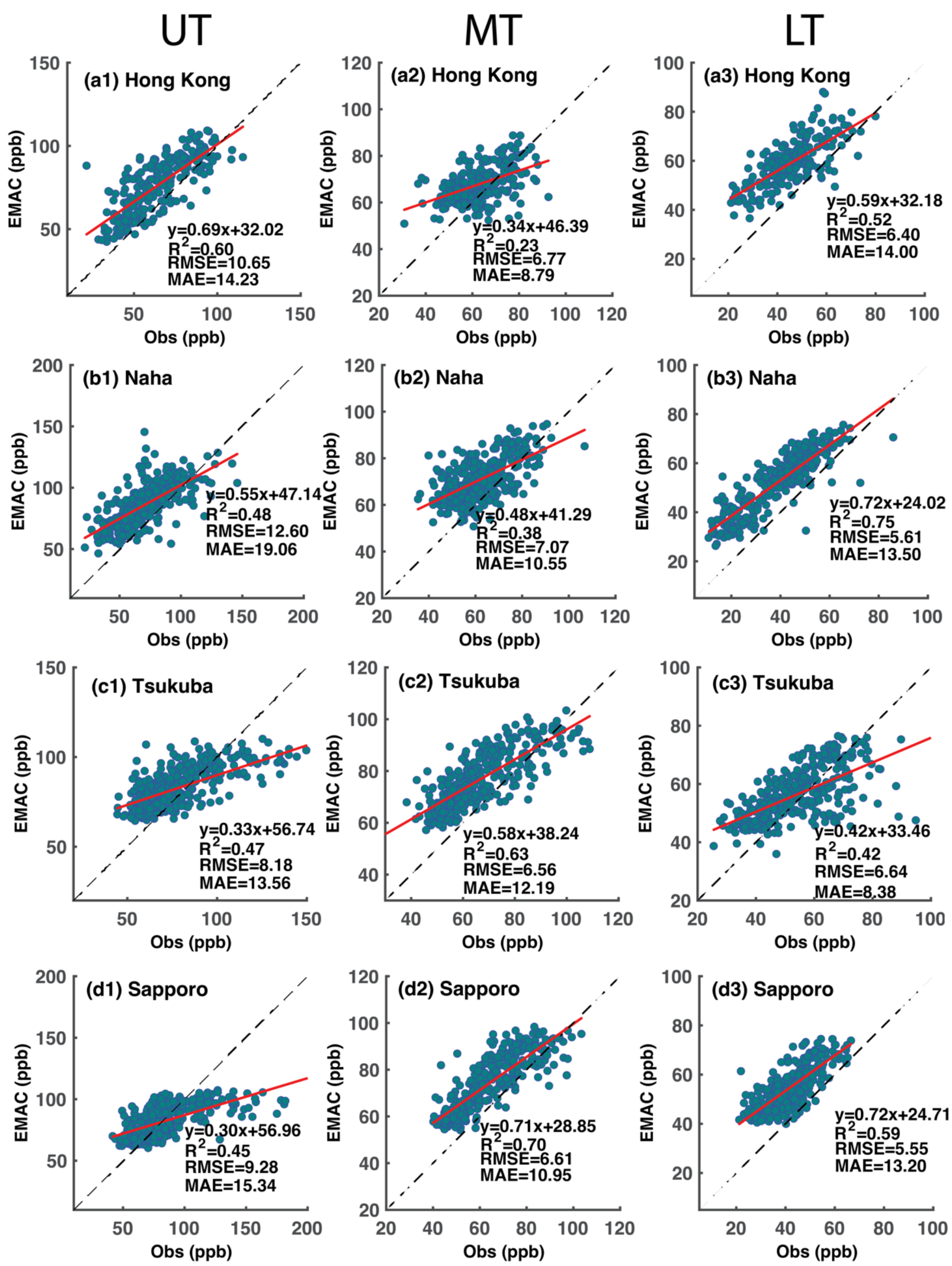

**Figure 6 Evaluation of $O_3$ simulated with the EMAC model with observations in the upper troposphere (UT), middle troposphere (MT), and lower troposphere (LT) at the four sites: (a1-c1) Hong Kong, (a2-c2) Naha, (a3-c3) Tsukuba, and (a4-c4) Sapporo. The red lines are linear regression results between the observations and the EMAC model results. Black dash lines are 1:1 for reference. The statistical metrics including the coefficient of determinations (R2), root mean standard error (RMSE), and mean absolute error (MAE) are included for the quantitative evaluation of the model performance.**

Furthermore, the EMAC model predicts the realistic long-term trends of $O_3$ at different levels of the troposphere as indicated by the similar $O_3$ changes between observation and model (Figure .7) as well as the comparable long-term change rates of model-predicted $O_3$ with the observations (Table 2). For example, the largest positive $O_3$ trends in the model also occur in the upper troposphere over Naha during summer at 0.75 ppb $a^{-1}$, slightly less than the observations with 0.82 ppb $a^{-1}$ for the past three decades (Table 2). Except for Hong Kong, the other three sites in the north have larger positive trends of $O_3$ in the upper troposphere than in the middle and lower troposphere from spring to autumn. Hong Kong shows a relatively large positive trend of $O_3$ in the middle and lower troposphere than other sites during the past 30 years.

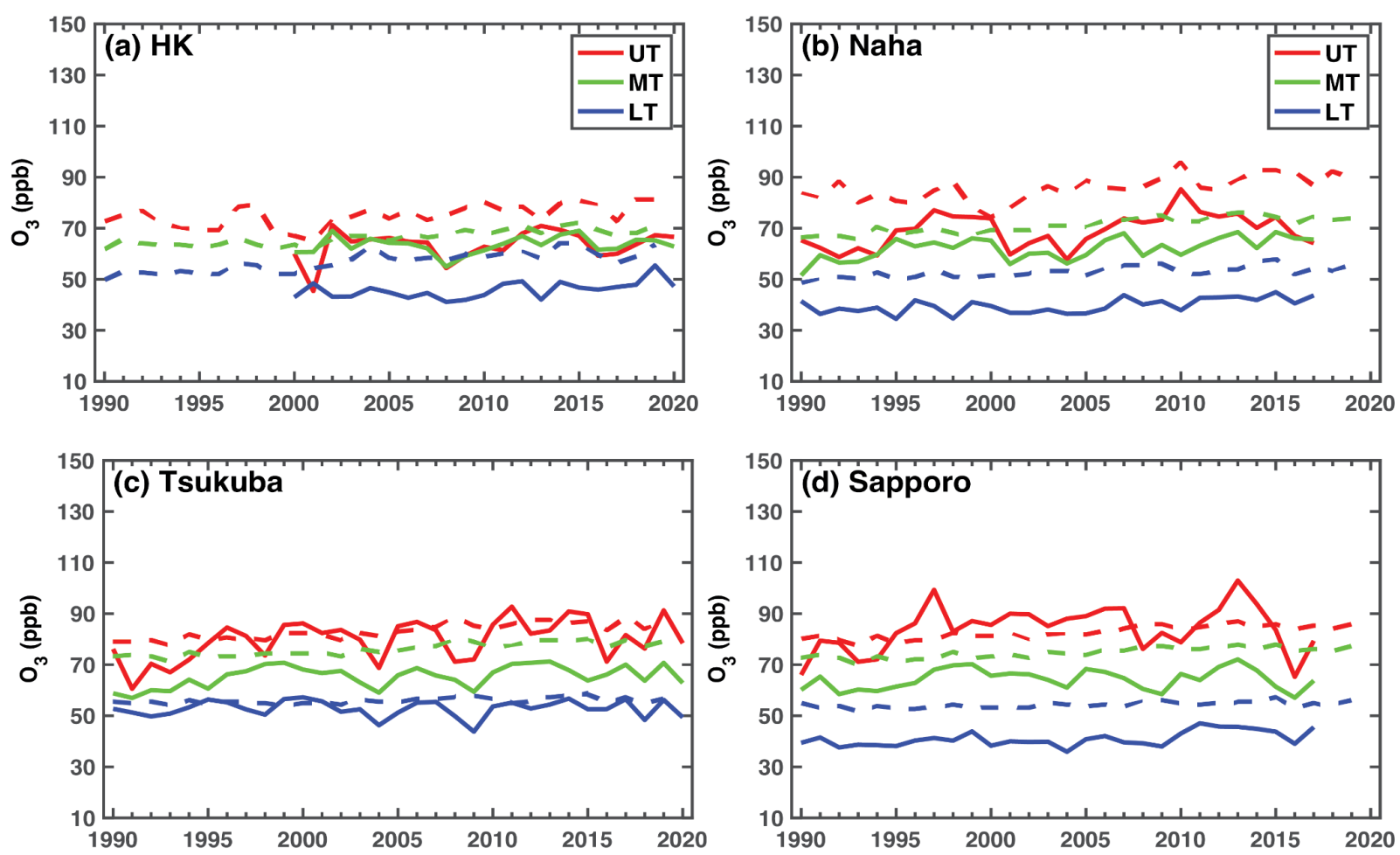

**Figure 7. Time series of ozone in ozonesonde (solid lines) and EMAC model (dash lines) for four sites at different layers of the troposphere.**

**Table 2. The trends of EMAC-simulated $O_3$ (ppb $a^{-1}$) in the upper, middle, and lower troposphere in different seasons from 1990 to 2019. The observational ozone trends are indicated in parentheses for comparison for the three Japanese sites. For the Hong Kong site, the $O_3$ trends since 2000 for both model (the first value) and observations (the second value) are in the square bracket. Note that observational periods for three Japanese sites are slightly different from the model (See Table 1). The trends with symbols (*) indicate the 95% confidence level. Bold indicates the agreement with the observations for significance and the sign of the trend, normal font for the sign of the trend but not for significance, and italic for the opposite sign of the trend.**

<table>
<tr><th>Station</th><th></th><th>MAM</th><th>JJA</th><th>SON</th><th>DJF</th></tr>
<tr><td rowspan="3">Hong Kong</td><td>UT</td><td>0.49* [0.98*|0.60*]</td><td>0.56* [0.49*|−0.15]</td><td>0.32* [0.34|−0.14]</td><td>0.06 [0.25|−0.07]</td></tr>
<tr><td>MT</td><td>0.33* [0.65*|0.20]</td><td>0.43* [0.39*|−0.11]</td><td>0.36* [0.29|0.01]</td><td>0.01 [−0.01|0.27]</td></tr>
<tr><td>LT</td><td>0.49* [0.65*|0.44*]</td><td>0.56* [0.53*|0.26]</td><td>0.32* [0.16|0.37]</td><td>0.06 [−0.18|0.16]</td></tr>
<tr><td rowspan="3">Naha</td><td>UT</td><td>0.33* (0.26)</td><td>0.75* (0.82*)</td><td>0.37* (0.21)</td><td>0.05 (0.06)</td></tr>
<tr><td>MT</td><td>0.42* (0.41*)</td><td>0.33* (0.44*)</td><td>0.33* (0.31*)</td><td>0.10* (0.02)</td></tr>
<tr><td>LT</td><td>0.32* (0.38*)</td><td>0.21* (0.23*)</td><td>0.09 (0.13)</td><td>0.08* (0.14*)</td></tr>
<tr><td rowspan="3">Tsukuba</td><td>UT</td><td>0.26* (0.63*)</td><td>0.45* (0.67*)</td><td>0.32* (0.31*)</td><td>0.12 (−0.05)</td></tr>
<tr><td>MT</td><td>0.21* (0.17)</td><td>0.37* (0.42*)</td><td>0.28* (0.29*)</td><td>0.14* (0.03)</td></tr>
<tr><td>LT</td><td>0.13*(0.11)</td><td>0.09 (−0.20)</td><td>0.03 (0.01)</td><td>0.05* (0.13*)</td></tr>
<tr><td rowspan="3">Sapporo</td><td>UT</td><td>0.22* (0.26)</td><td>0.34* (0.68)</td><td>0.25* (0.13)</td><td>0.15* (0.07)</td></tr>
<tr><td>MT</td><td>0.18* (0.12)</td><td>0.28* (0.14)</td><td>0.21* (0.04)</td><td>0.11* (0.06)</td></tr>
<tr><td>LT</td><td>0.12* (0.29*)</td><td>0.12* (0.05)</td><td>0.03 (0.13)</td><td>0.03 (0.31*)</td></tr>
</table>

Figure 8 demonstrates the month-height cross-sections of EMAC-predicted monthly-mean $O_3$ and their changes in the troposphere at the four sites between the 1990s and 2010s. Compared with the observed counterparts (Figure 3), the model reproduces the temporal-spatial variation patterns of tropospheric $O_3$ within the troposphere quantitatively well. Specifically, the model captures a key feature with the ozone tongue that occurs from late spring to early summer over four sites and their variation with latitude. The summer relatively "clean" layer with low $O_3$ mixing ratios in the lower troposphere at the southern sites of Hong Kong and Naha is also well simulated.

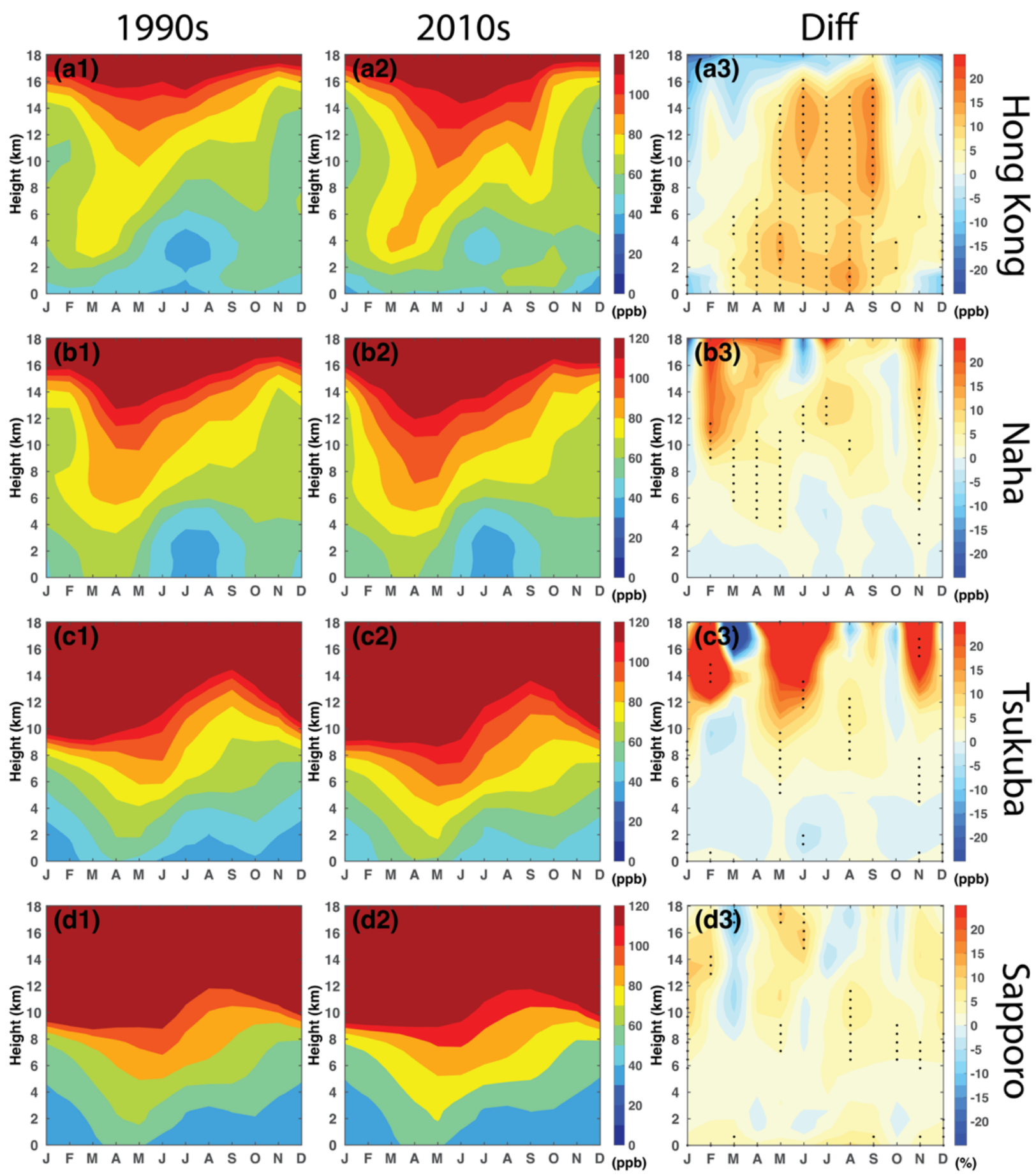

**Figure 8. EMAC-simulated monthly mean $O_3$ in the 1990s and 2010s, and their differences between 2010s and 1990s at the four observation sites (a1-a3) Hong Kong, (b1-b3) Naha, (c1-c3) Tsukuba and (d1-d3) Sapporo. The horizontal axes denote the months of the year and the vertical axes represent the height above ground. Dots in the i-l represent the layer with statistically significant changes according to a paired two-sided t-test ($p < 0.05$).**

Overall, the EMAC model reasonably simulates the spatial and temporal variations in tropospheric $O_3$ as compared to the $O_3$ observations at the four sounding sites. Consistency between the model and observations suggests that the trends observed in the Japanese ozonesondes remain valuable despite uncertainties related to the transitions between the two types of ozonesondes. Moreover, the model can effectively be used to investigate the drivers of these trends.

### 3.2.2 Changes in $O_3S$ and $O_3T$ derived from EMAC simulations

To gain deeper insights into the factors contributing to tropospheric $O_3$, we analyze the EMAC-simulated total $O_3$ in the troposphere, origin of $O_3$ from the stratosphere (i.e., stratospheric intrusion, $O_3S$), and origin of $O_3$ from the troposphere (i.e., photochemical production in the troposphere, $O_3T$) at the four sites, along with their latitudinal variations (Figures 9 and 10). The layer with the large mixing ratio of $O_3S$ extending from the lower stratosphere to the troposphere occurs in early spring in the southern site (i.e., Hong Kong). Conversely, similar occurrences are observed to shift to early summer in the northern site (i.e., Sapporo) (Figure 9). The seasonal buildup of mid-latitude total $O_3$ typically unfolds from winter through late spring, followed by a decline in summer (Fioletov and Shepherd, 2003). Furthermore, together with dynamical processes such as tropopause folding in the vicinity of the subtropical jet (Baray et al., 2000), stratospheric $O_3$ is transported downward into the troposphere. Over the past 30 years, the two sites within the subtropics (Tsukuba at 36ºN and Sapporo at 43ºN) exhibit larger $O_3S$ increases in the lower stratosphere and upper troposphere compared to the other two sites situated in the near-tropical region (Hong Kong at 22ºN and Naha at 26ºN).

The $O_3T$ shows seasonal maxima during the warm seasons (from March to October) throughout the troposphere in Hong Kong, while mainly occurring in the middle to upper troposphere among three Japan sites (Figure 10). In the lower troposphere at Hong Kong, the $O_3T$ contributes more than $O_3s$ (60-80 ppb vs. 10-20 ppb) in the separated $O_3$ hotspots around 2-4km during spring. In the tropical regions, air rises in the Hadley cell from the surface to the upper troposphere, and further ascent into the stratosphere where it is transported to the mid-latitudes by the Brewer-Dobson Circulation (Brewer, 1949; A. Stohl et al., 2003). In this way, the tropospheric origin $O_3$ could be further transported to the middle-upper troposphere of middle-latitude regions.

Several factors influence $O_3$ mixing ratios over study regions, which could potentially be responsible for the local maxima in $O_3T$: transport from near-surface tropospheric $O_3$ within the upward branch of the Hadley cell into the upper troposphere; horizontal transport from upstream polluted regions, e.g., mainland China in this study; biomass burning related transport; enhanced mixture by active convection and lighting events; local photochemical $O_3$ production. $O_3T$ has shown significant enhancements among the four sites over the past several decades. However, the primary contributors to the high $O_3T$ concentrations and their enhancement vary with locations and layers, which require further investigation.

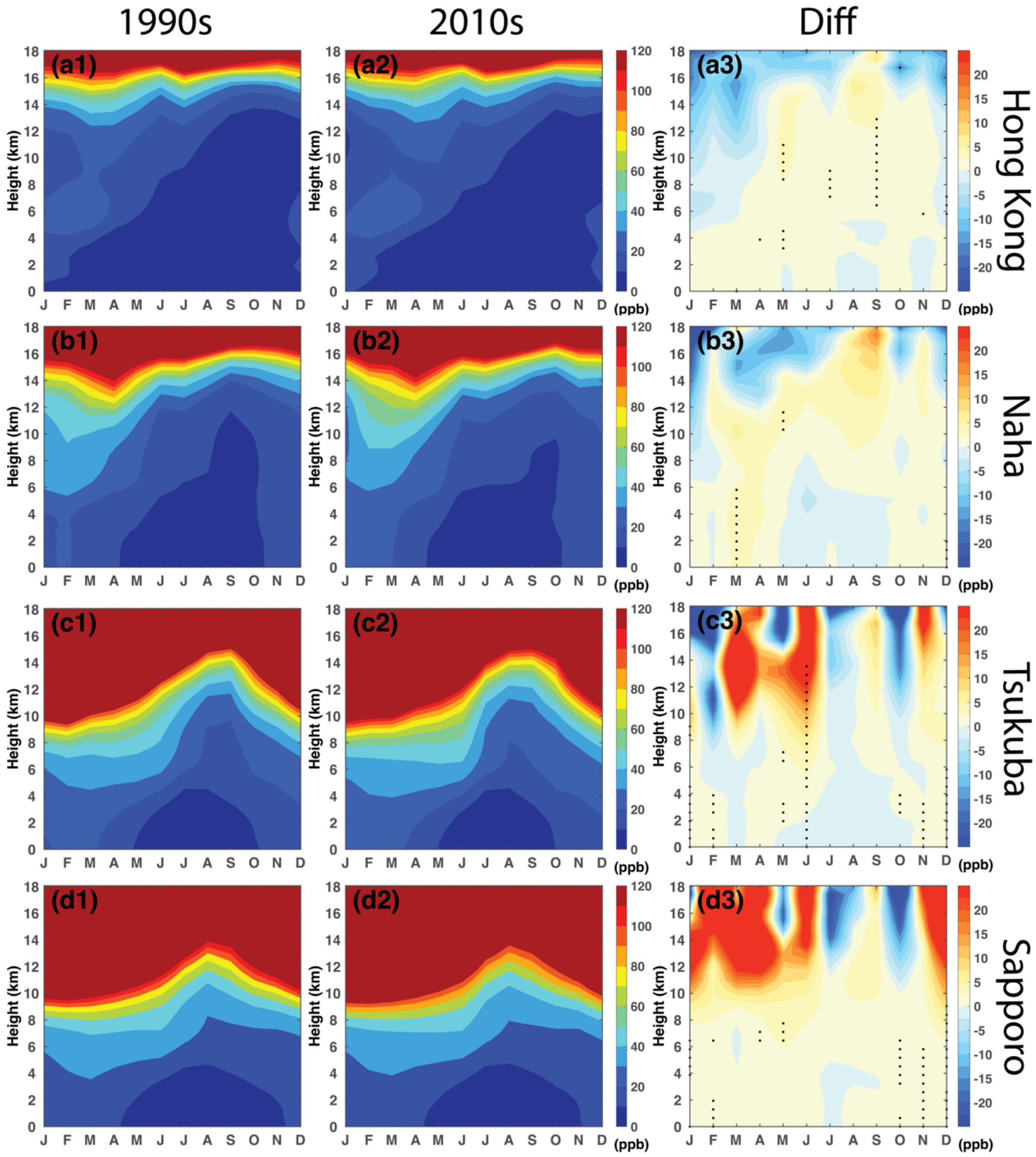

Figure 9. A comparison of the EMAC-simulated monthly mean temporal and spatial distributions of $O_3S$ in the 1990s and 2010s, and the difference between 2010s and 1990s at the four observation sites: Hong Kong, Naha, Tsukuba, and Sapporo. Dots represent the layer with statistically significant changes according to a paired two-sided t-test ($p < 0.05$).

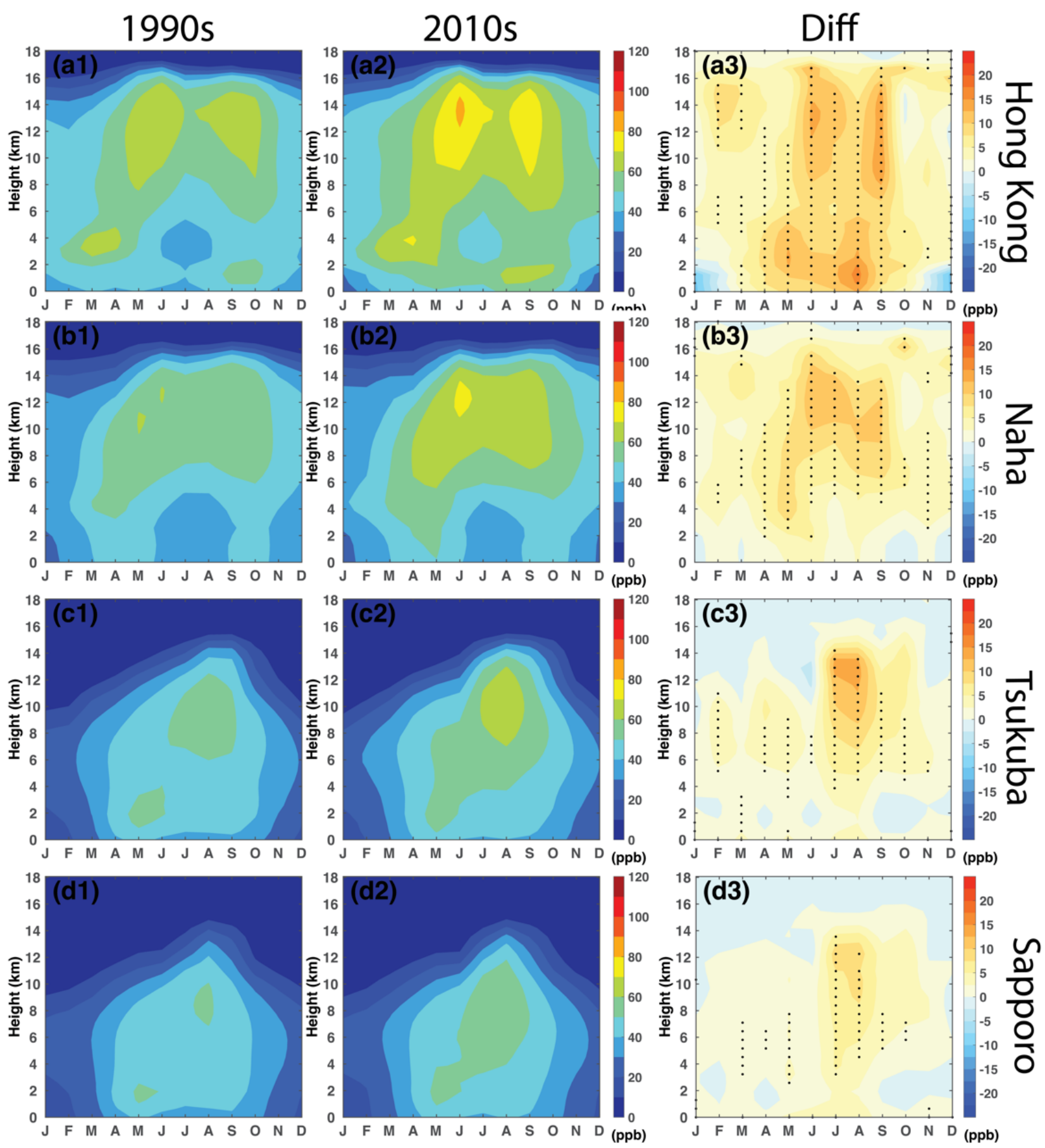

**Figure 10. Similar to Figure 9 but for the component of tropospheric $O_3$ ($O_3T$).**

### 3.2.3 Quantification of stratospheric intrusion vs. tropospheric production using EMAC

Utilizing the reasonably realistic simulations of tropospheric $O_3$ and their variations by the EMAC model, we can now quantify the respective contributions of $O_3S$ and $O_3T$ to the changes in tropospheric $O_3$ between the 2010s and 1990s, as presented in Table 3. Overall, the increase of $O_3T$ (up to 11.09 ppb) dominates the $O_3$ increase throughout the troposphere at all the sites during summer. Particularly for the near-tropical sites, Hong Kong and Naha, the increase of $O_3T$ contributes more than the $O_3S$ changes with percentage contributions greatly much more than 60%, even offsetting the decrease in $O_3S$ during winter and spring. Conversely, for the subtropical sites, Tsukuba and Sapporo, $O_3S$ emerges as the primary driver for changes in the middle-upper tropospheric $O_3$ during winter and spring. The contribution of $O_3S$ to observed $O_3$ increases by up to 96% and 40% in the middle-upper troposphere during winter and summer.

**Table 3. Contribution from $O_3S$ and $O_3T$ to changes of tropospheric $O_3$ between the 2010s and 1990s at the upper, middle, and lower troposphere (UT, MT, and LT) in different seasons. The percentage contributions of $O_3S$ and $O_3T$ to $O_3$ changes are listed in the parentheses.**

| Station | | $O_3S$ changes (ppb) | | | | $O_3T$ changes (ppb) | | | |
|---|---|---|---|---|---|---|---|---|---|
| | | MAM | JJA | SON | DJF | MAM | JJA | SON | DJF |
| Hong Kong | UT | −2.03 (−57%) | 1.44 (11%) | 1.41 (20%) | −3.44 (860%) | 5.58 (157%) | 11.09 (89%) | 5.69 (80%) | 3.04 (−760%) |
| | MT | 1.30 (20%) | 0.96 (10%) | 1.23 (16%) | −2.84 (−888%) | 5.06 (80%) | 8.27 (90%) | 6.27 (84%) | 3.16 (988%) |
| | LT | 0.88 (9%) | 0.10 (1%) | −0.13 (−2%) | 1.24 (59%) | 8.73 (91%) | 11.37 (99%) | 6.41 (102%) | 0.86 (41%) |
| Naha | UT | 1.05 (18%) | 3.81(26%) | 2.98 (38%) | −1.87 (−143%) | 4.90 (82%) | 10.95 (74%) | 4.78 (62%) | 3.18 (243%) |
| | MT | 2.32 (27%) | 0.08 (1%) | 1.10 (16%) | −1.03 (−47%) | 6.19 (73%) | 6.22 (99%) | 5.64 (84%) | 3.22 (147%) |
| | LT | 2.35 (40%) | −0.19 (−6%) | 0.07 (4%) | 0.73 (43%) | 3.51 (60%) | 3.51 (106%) | 1.68 (96%) | 0.98 (57%) |
| Tsukuba | UT | 7.33 (69%) | 4.23 (40%) | 2.19 (34%) | −4.59 (221%) | 3.32 (31%) | 7.22 (60%) | 4.15 (66%) | 2.51 (−121%) |
| | MT | 1.50 (33%) | 2.10 (28%) | 1.39 (27%) | 0.51 (19%) | 3.04 (67%) | 5.29 (72%) | 3.79 (73%) | 2.23 (81%) |
| | LT | 1.27 (51%) | 0.44 (20%) | 0.94 (392%) | 0.90 (92%) | 1.22 (49%) | 1.74 (80%) | −0.70 (−292%) | 0.08 (8%) |
| Sapporo | UT | 6.85 (79%) | 3.19 (37%) | 2.00 (39%) | 4.65 (96%) | 1.82 (21%) | 5.40 (63%) | 3.11 (61%) | 0.17 (4%) |
| | MT | 1.60 (42%) | 1.59 (28%) | 1.31 (34%) | 1.62 (71%) | 2.20 (58%) | 4.14 (72%) | 2.57 (66%) | 0.65 (29%) |
| | LT | 1.19 (50%) | 0.35 (13%) | 0.71 (263%) | 0.69 (115%) | 1.18 (50%) | 2.45 (87%) | −0.45 (−163%) | −0.09 (−15%) |

To get a more complete picture of how tropospheric $O_3$ changes along the Northwest Pacific regions, the zonal mean of tropospheric $O_3$, $O_3S$, and $O_3T$ changes are compared in Figure 11. The climatological distribution of vertical tropospheric $O_3$ with latitude is determined by $O_3S$ in the subtropics and $O_3T$ in the tropics.

Tropospheric $O_3$ shows statistically significant positive changes from 10ºN to 60ºN in summer, with the maximum in the middle to upper troposphere around 30ºN. Similarly, $O_3T$ demonstrates a similar pattern of changes as tropospheric $O_3$ in summer, indicating that tropospheric photochemical $O_3$ production is the primary driver of the summertime tropospheric $O_3$ enhancement. Strengthened downward transport of stratospheric $O_3$ primarily affects the upper troposphere in the subtropics during summer.

Conversely, during winter and spring, the $O_3S$ significantly contributes to the enhancement of tropospheric $O_3$ in the subtropics. Positive changes in $O_3T$ are observed south of 40ºN, partly offsetting the decrease in $O_3S$ in the upper troposphere.

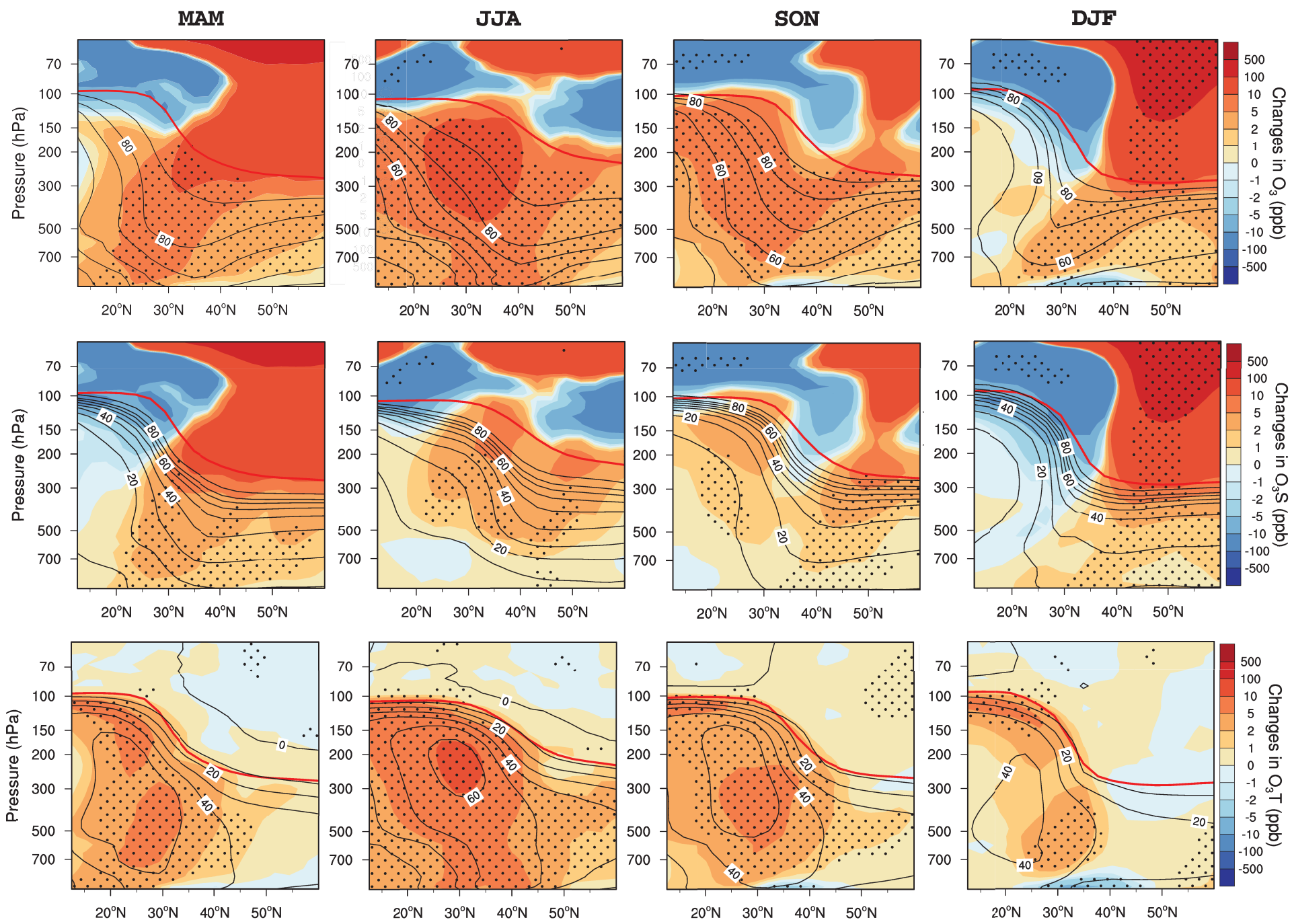

**Figure 11. Latitude-pressure cross sections of mixing ratio difference of $O_3$, $O_3S$, and $O_3T$ (ppb) between the 2010s and 1990s along the Northwest Pacific region (zonal mean over 110°N to150°N) in four seasons. Black lines indicate the climatological distribution. Red solid lines denote the tropopause height. Dots represent the layer with statistically significant changes according to a paired two-sided t-test ($p < 0.05$).**

## 4. Discussion and Conclusion

In this study, thirty years of ozonesonde observational data at four ozonesonde sites (Hong Kong, Naha, Tsukuba, and Sapporo) are presented together with simulation results of the chemistry-climate model EMAC to characterize the temporal and spatial variation patterns and the long-term changes of tropospheric $O_3$ along the Northwest Pacific region.

The analysis of the seasonality in $O_3$ shows a seasonal maximum throughout the troposphere, occurring in late spring at the tropical site Hong Kong and shifting to early summer at the mid-latitude sites like Sapporo. Additionally, for Hong Kong and Naha, the lower tropospheric $O_3$ exhibits a seasonal minimum. As for long-term changes, tropospheric $O_3$ generally increases at all four sites. Naha and Tsukuba, show larger positive trends of $O_3$ up to 0.82 ppb $a^{-1}$, particularly in the upper and middle troposphere. The aggregation analysis between different decades indicates that the seasonal maximum in the troposphere becomes more pronounced and deeper over time.

Based on EMAC simulations, the summer and autumn enhancement of $O_3$ in the middle-upper troposphere is mostly attributable to tropospheric ozone source linked to increasing pollution emissions, with percentage contributions more than 60%. On the other hand, ozone originating from the stratosphere dominates the large

portion of middle-upper tropospheric $O_3$ enhancement by up to 96% and 40% in the mid-latitude during winter and spring. The climatological maximum observed in the seasonality of ozone throughout the troposphere is associated with both stratosphere-troposphere exchange north of 30°N and photochemical $O_3$ production in the troposphere in spring. These findings corroborate the features discussed by Oltmans et al. (2004), confirming them with a longer observational dataset based on the tagged ozone tracers in the EMAC model. Our results further confirm the offsetting effect of $O_3T$ increase to the decrease in $O_3S$ in the tropical troposphere during winter and spring.

While the magnitude of $O_3$ trends is well simulated with the EMAC model in most atmospheric layers, uncertainties persist in the mean values due to various factors. These include large dynamical variability perturbing stratosphere-to-troposphere $O_3$ transport, the influence of $O_3$-depleting substances, uncertainties of long-term changes in emissions, insufficient treatment of chemical processes, or inaccurate transport due to excessive numerical diffusion in the tropopause region, etc. Additionally, uncertainties may arise from interpolating the relatively coarse horizontal and vertical resolution of the global model data to the locations of the observational sites. Nevertheless, the presented results indicate a satisfactory level of agreement between the model results and the observations, allowing further disentangling of $O_3T$ versus $O_3S$ contributions.

The dynamical and chemical drivers for such long-term tropospheric changes deserve further analysis in the future. Here, we propose some mechanisms based on related research that could potentially contribute to observational tropospheric $O_3$ enhancements in East Asia. Regional transport is one important contributor to tropospheric $O_3$ enhancement. Compared with the other two Japanese sites, Naha, to the east of China, is susceptible to regional transport of air pollution from China. The prevailing westerly winds bring $O_3$-enriched air from eastern China to Naha, resulting in a substantial increase of $O_3$ from the middle to upper troposphere. Internal dynamical variabilities such as the warm phase of El Niño-Southern Oscillation (ENSO) and the easterly phase of the Quasi-Biennial Oscillation (QBO) are known to be closely tied to enhanced STT of $O_3$ (Neu et al 2014, Zeng and Pyle, 2005). The ENSO/QBO-related changes can influence jet stream variations, leading to the formation of tropopause folds through Rossby wave breaking (Albers et al 2018). Increased frequency and the northward shift of tropopause folding events are observed in the East Asia region (Figure S3), attributed to an increase in the zonal wind and poleward-upward shift of the STJ driven by global warming–induced increases in greenhouse gasses (Akritidis et al 2019, Manney and Hegglin, 2018). With increasing greenhouse gasses, the Brewer-Dobson circulation tends to strengthen due to larger zonal-mean temperature gradients and increased wave drag in the extratropical stratosphere (Shepherd and McLandress, 2011; Neu et al 2014). This results in an increased $O_3$ reservoir over the subtropical LMS, facilitating downward transport to the troposphere u under the influence of the Pacific jet (Hegglin and Shepherd, 2009; Albers et al 2018).

**Data Availability Statement**: The ozone-sounding dataset used for observational analysis in the study is publicly available at the World Ozone and Ultraviolet Radiation Data Centre via https://woudc.org/data/explore.php?lang=en (last access: 25 Feb 2024).

**Supplement:** Supplementary.pdf

**Author Contributions:** XM carried out all the observational and model simulation data analyses, led the interpretation of the results, and prepared the manuscript with contributions from all the co-authors. JH, MH, PJ, and TZ contributed to the interpretation of the results and provided extensive comments on the manuscript. PJ conducted the EMAC simulations.

**Competing interests:** At least one of the (co-)authors is a member of the editorial board of Atmospheric Chemistry and Physics.

**Acknowledgment:** This research has been supported by the National Key Research and Development Program of China (2022YFC3701204), the National Natural Science Foundation of China (42275196, 42105164), and the Applied Basic Research Foundation (2022A1515011078). The EMAC simulations have been performed at the German Climate Computing Centre (DKRZ) through support from the Bundesministerium für Bildung und Forschung (BMBF). DKRZ and its scientific steering committee are gratefully acknowledged for providing the HPC and data archiving resources for this consortial project ESCiMo (Earth System Chemistry integrated Modelling). We especially thank the Michael Sprenger from ETH Zurich for providing the tropopause folding frequency dataset.

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
