# Peer review of "Causes of growing middle-upper tropospheric ozone over the 1"

_EGUsphere, 2023_

## Referee Comment (RC2)

**Review of "Causes of growing middle-upper tropospheric ozone over the North West Pacific region" by Xiaodan Ma et al.**

**Overall Assessment**

This study leverages ozonesonde records over a relatively long (>20 year) period for four locations bordering the North-West Pacific Ocean, in comparing a state-of-the-art chemistry climate model (EMAC) in simulating ozone across different latitudes spanning the near-tropics to mid-latitudes, as well as from the surface to the upper-troposphere-lower stratosphere region. The authors confirm good overall agreement in terms of EMAC's ability to capture the seasonality in tropospheric ozone, as well as in many cases relatively large, robust trends/changes that have occurred at each location from the 1990s to 2010s, according to the ozonesonde records. They do this by separating out upper, middle and lower troposphere, abbreviated UT, MT and LT respectively, and computing monthly/seasonal means. By exploiting the stratosphere-tagged ozone tracer ($O_3S$) in the EMAC model, the authors then proceed to quantify the attribution from the stratosphere versus that formed in the troposphere according to the model climatology. A detailed assessment of the stratospheric contribution to recent trends/changes is then performed. Calculation of the residual amount of ozone formed in the troposphere ($O_3T$) helps to then attribute the role of tropospheric production.

The work is of clear importance as ozone is a non-well mixed greenhouse gas, with a typical lifetime of ~3 weeks in the free troposphere, and is sensitive to multiple factors, including in situ photochemical production, long-range transport and large-scale dynamics. Uncertainty in what drives/influences tropospheric ozone is therefore large and there is an obvious need to understand this better as there is an obvious radiative forcing and air quality implication to the results, as recognised by the wider literature. Dynamical influence mainly entails changes in the downward transport of ozone from the stratosphere, which is strongly determined by the strength of the Brewer-Dobson Circulation (the residual meridional stratospheric overturning circulation forced by wave-mean flow interactions). Facilitation of such downward transport arises from the action of tropopause folding, which preferentially occurs in association with the subtropical/eddy-driven jets and transient synoptic-scale systems (e.g., cut-off lows). Filamentation of ozone-rich air near the tropopause can subsequently be entrained into the troposphere (so-called 'stratospheric intrusions').

This regional-focussed study is therefore highly valuable and is complementary to many earlier studies which have a more general focus (e.g., global). The study is logically presented and well executed, fitting well within the scope of Atmospheric, Chemistry and Physics journal. I would recommend publication of the study, following a minor round of revisions as per my general and minor comments which I detail below:

**General Comments**

**L56:** "…Brewer-Dobson circulation promotes stratospheric intrusions…" → I wouldn't say this is true. An enhanced BDC may facilitate an enrichment of ozone into the extratropical lowermost stratosphere that is readily available to be transported down into the troposphere. The development of stratospheric intrusions is instead governed by processes operating on much finer spatial scales,

typically synoptic-scale or less, such as associated with the rear-flank of extratropical cyclones/cut-off lows. The authors detail this later (L63-65).

The authors I think need to change the wording to reflect that an increase in BDC strength can promote greater seasonal build-up of ozone into the extratropical lowermost stratosphere over winter (Ray et al., 1999; Konopka et al., 2015; Ploeger & Birner, 2016). Subsequent stratospheric intrusions can them facilitate greater stratosphere-troposphere exchange of ozone as a result of this enrichment, particularly around March time when the lowermost stratospheric reservoir of ozone reaches an annual maximum and is seasonally "flushed thereafter" (Hegglin and Shepherd, 2007; Bönisch et al., 2009). This however does depend on changes to each of the deep and shallow branches of the BDC; a strengthening of the deep branch indeed serves to increase lowermost stratospheric ozone. Strengthening of the shallow branch on the other hand favours enhanced transport/mixing of low-ozone air from the tropical upper troposphere (e.g., Fig. 4 in Bönisch et al., 2009).

Given the BDC is pertinent to this study, I suggest the authors also introduce this key distinction between the shallow and deep branches and their significance in regard to stratosphere-troposphere exchange of ozone.

**L208-209:** "The tropopause folding on the south part of the STJ could lead to more stratospheric intrusion contributions to the ozone tongue." → I think it would help if the authors provided an overlay in Figure 2 of the standard deviation or similar (to highlight interannual variability). Not only because this could provide more useful information, but it would be expected I think that higher standard deviation would support this speculative point. I wonder if the authors could also go a step further and attempt to diagnose the STJ from one year to the next, which would bolster this assertion?

**Sect 3.2.1 (L337-346):** Although EMAC generally overestimates ozone, there is clearly a tendency towards higher overestimation for lower ozone mixing ratios and lower overestimation at higher ozone mixing ratios (which changes in sign for most distributions, such that EMAC typically underestimates ozone at the highest ozone mixing ratios). The authors neglect to mention this so should consider adding description/quantification of this.

I also think the authors need to be a little bit careful with the interpretation for instance that EMAC better represents the upper and lower troposphere versus the middle troposphere for Hong Kong/Naha. The RMSE and MAE imply otherwise at face value, though this is because the differences are less systematic and thus compensated for. The total error summed for all soundings versus model values would maybe be larger for the MT but hard to tell as the axes' ranges are smaller for MT and LT compared with UT. Perhaps the authors would like to compute this, add and compare, which would help support their claim?

**Minor Comments**

**L46:** "Tropospheric $O_3$ increases of 7% (measured as a partial column between 3-9 km)…" → The authors may wish to specify this increase is of "Free tropospheric $O_3$", thereby excluding potentially different trends in the lower troposphere/ABL.

**L51-52:** "…, the counterpart in the middle-upper troposphere…" → "…, the evolution in the middle-upper troposphere…"?

**L58:** "a stratospheric chemistry-climate model" → Is this correct or should this be a chemistry-climate model which has a well-resolved stratosphere?

**L118-120:** "The research from the cross-evaluation of OMI data and the ozonesonde observation in Japan sites shows that CI ozonesonde measurements are negatively biased relative to ECC measurements by 2–4 DU compared with the OMI data (Bak et al., 2019)". → Unclear if the bias of 2-4 DU is for total column, or partial column only in troposphere.

**L199:** "Figure 2 depicts the climatologically vertically resolved tropospheric O3 distribution with respect to months" → Suggested sentence revision: "Figure 2 depicts the monthly climatological vertically resolved tropospheric $O_3$ distribution throughout the year".

**Figure 2:** Yellow colour shading missing for Tsukuba. This needs checking.

**Figure S1:** Same as issue as above for panels c1 (Tsukuba) and d1-d2 (Sapporo).

**L208:** "Figure S2" → Change to Figure S1 and reverse order of existing Figures S1 and S2 so that they are in order.

**Figure 3 Caption:** I think it would be helpful to remind the reader of what is used to represent upper, middle and lower troposphere. I'm not entirely convinced that each should be best represented by a single pressure level though, particularly for the lower troposphere where ABL and free tropospheric ozone amounts and variability (including its trends) can differ substantially. Also, are the time periods the same as in Figure 2?

**Figure 4c1 and 4d1-2:** Same issue as before with contour shading, certain colours missing.

**Figure 4 Caption**: "Dashed lines in the i-l represent…" → Typo? I assume this refers to the c panels for each site?

**L280:** "Figure S1" → See L208 comment.

**L287:** "20 to 40 ppb." → Difficult to tell as colour scale saturates above 20 ppb? Should this be extended?

**L288:** "…Hong Kong shows more significant O3 changes in the lower troposphere" → This seems true in the ABL (~<1.6 km) and above what is likely 850 hPa (which I think the authors still use to represent the lower troposphere) for April and August (~2-4 km). However, there is a region in

between which is statistically insignificant throughout the calendar year. I don't feel the text gives full justice to the nature of the results shown in Figure 4a3.

**L333:** "3.2 Quantification of stratospheric intrusion versus tropospheric production using EMAC simulations" → It seems only later subsection 3.2.3 directly relates to this heading (the later sub-heading is essentially worded the same).

May I suggest to the authors re-writing the section 3.2 header to encompass the model-measurement evaluation of tropospheric ozone first in 3.2.1, something along the lines of: "3.2 Comparison with observations and stratospheric versus tropospheric attribution using EMAC simulations"

**L344:** "Figure 7c2" → I think this refers to Figure 6c2.

**L345:** "Figure 7 b1" → I think this refers to Figure 6b1.

**L356:** "Figure .7" → Figure 7

**L375-376:** "Bold indicates the agreement with the observations for significance and the sign of the trend" → Not all bold entries have asterisks next to both the model and observed trend. Is there a mistake or did I misinterpret the description?

**L388:** "The EMAC simulations of O3 at different portions of the troposphere…" → Revised sentence suggestion: "The EMAC simulations of O3 for different altitude ranges in the troposphere…"

**L389:** "Dots in the i-l represent…" → Again, not clear what i-l stands for. Is it a typo?

**Figure 8d1-d2** → Check colour scale. Seems that yellow filled contour is missing in Figure 8d1 and possibly cyan filled contour missing in both.

**L406-407:** "Furthermore, together with dynamical processes such as tropopause folding in the vicinity of the subtropical jet (Baray et al., 2000), stratospheric $O_3$ is transported downward into the troposphere." → The authors should also add mention of the seasonal lifting of the which will naturally contribute to entrainment of ozone-rich air from the stratosphere into the troposphere (e.g., Monks, 2000).

**Figure 9d2** → Yellow colour missing.

**L434:** "3.2.3 Quantification of stratospheric intrusion vs. tropospheric production using EMAC" → This is almost word for word identical to the 3.2 heading on L333. It is more fitting here, but I would simplify by heading it as follows: "3.2.3 Attribution of EMAC tropospheric O3 changes: $O_3S$ vs. $O_3T$"

**L442-443:** "The contribution of O3S to observed O3 increases by up to 96% and 40% in the middle-upper troposphere during winter and summer." → I'm a little unclear where these numbers are from. Are they directly shown in Table 3? 96 % is shown for Sapporo in JJA and 40 % for Tsukuba in

JJA (UT), could that be what the authors refer to? It needs to be clearer exactly what season, site and region of troposphere this is (the authors state middle-upper troposphere so reading this implies the quantified contribution is a conflation of the two regions).

**Table 3:** It seems to me there is a missing set of numbers and that is the total change in O₃ between the 2010s and 1990s for the EMAC model (noting that similar such values in Table 2 are linear trend changes and not the difference between the later and earlier period). These numbers would be good to have so the reader can fully appreciate attribution from the stratosphere versus troposphere without having to do the math.

There appears to be a few inconsistencies in the Table. Presumably the absolute and percentage change values should be equivalent in sign for each season, location and each of UT, MT and LT, which is not always the case. This need checking.

**L469:** "110°N to150°N" → "110°E to 150°E"

**Figure 11:** I wonder if the authors could add an equivalent Figure but expressed as a percentage (this could go in the supplementary information at the end). This would be extremely handy to compare and should be minimal effort for the authors to produce and include.

**L488-490:** "On the other hand, ozone originating from the stratosphere dominates the large portion of middle-upper tropospheric O3 enhancement by up to 96% and 40% in the mid-latitude during winter and spring. → Again, these numbers are very specific and I'm not sure if they are plucked directly from Table 3 or not. If the latter, it is surely better to give a range that encompasses both MT and UT, including both Tsukuba and Sapporo as I think the authors are using to represent mid-latitudes.

**L518:** "With increasing greenhouse gasses…" → "With increasing greenhouse gases…"

**L521:** "…facilitating downward transport to the troposphere u under the influence of the Pacific jet…" → "…facilitating downward transport to the troposphere under the influence of the Pacific jet…"

**References**

Bönisch, H., Engel, A., Curtius, J., Birner, Th., and Hoor, P.: Quantifying transport into the lowermost stratosphere using simultaneous in-situ measurements of SF₆ and CO₂, Atmos. Chem. Phys., 9, 5905–5919, https://doi.org/10.5194/acp-9-5905-2009, 2009.

Hegglin, M. I., & Shepherd, T. G. (2007). O3-N2O correlations from the atmospheric chemistry experiment: Revisiting a diagnostic of transport and chemistry in the stratosphere. *Journal of Geophysical Research*, *112*, D19301. https://doi.org/10.1029/2006JD008281.

Konopka, P., Ploeger, F., Tao, M., Birner, T., & Riese, M. (2015). Hemispheric asymmetries and seasonality of mean age of air in the lower stratosphere: Deep versus shallow branch of the Brewer-

Dobson circulation. *Journal of Geophysical Research: Atmospheres*, *120*, 2053–2066. https://doi.org/10.1002/2014JD022429.

Monks, P. S. (2000). A review of the observations and origins of the spring ozone maximum. *Atmospheric environment*, *34*(21), 3545-3561, https://doi.org/10.1016/S1352-2310(00)00129-1.

Ploeger, F. and Birner, T.: Seasonal and inter-annual variability of lower stratospheric age of air spectra, Atmos. Chem. Phys., 16, 10195–10213, https://doi.org/10.5194/acp-16-10195-2016, 2016.

Ray, E. A., Moore, F. L., Elkins, J. W., Dutton, G. S., Fahey, D. W., Vomel, H.,…Rosenlof, K. H. (1999). Transport into the Northern Hemisphere lowermost stratosphere revealed by in situ tracer measurements. *Journal of Geophysical Research*, *104*, 26,565–26,580, https://doi.org/10.1029/1999JD900323.

---

## Author Response (AR1)

Dear Editors:

We are submitting the revised manuscript acp-2023-2411 together with point-by-point responses to the reviewer' comments. We appreciate the reviewers for the valuable comments. We hope you and the reviewers are satisfied with our responses and revision.

The major changes made in the revised version include:

1. Added discussion about how Brewer-Dobson circulation impacts stratospheric to tropospheric exchange in Introduction (L55-64).

2. Added Figures S1 and S2, as well as a discussion about the change of ozonesonde and its impact on the long-term ozone trends in the revised Supplementary material (L30-61).

3. Added more discussion about the evaluation of EMAC simulations in 3.2.1 (L355-368).

4. Added tropospheric ozone changes in Table 3 for better comparison (L462-464).

5. Submitted the EMAC model output used in the paper on Zenodo (https://zenodo.org/records/11093806 ); the corresponding information has been updated in the Data Availability Statement (L536-537).

6. Added S4 to investigate the spatial relation between the subtropical jet stream and tropopause folding events in the revised Supplementary material (L70-73).

We look forward to hearing from you regarding our revision soon.

On behalf of the co-authors,

First Author: Xiaodan Ma
Corresponding Author: Jianping Huang

**Point-by-point responses to the Review Comments #1**

*Review of "Causes of growing middle-upper tropospheric ozone over the Northwest Pacific region" by Xiaodan Ma et al. Summary and General Comments:*

*This paper presents an analysis of ozone trends from ozonesonde observations and EMAC model simulations to demonstrate positive ozone trends in the mid-to-upper troposphere over the past ~30 years. The authors analyze the seasonal and spatial ozone distribution from ozonesonde profiles at Hong Kong, Naha, Tsukuba, and Sapporo and leverage model ozone tracers to attribute tropospheric ozone growth primarily to production in the troposphere, with some input from increased stratosphere-to-troposphere transport.*

***General Comment:*** *The CI to ECC ozonesonde transition at the Japanese stations in ~2009 is mentioned, and the authors correctly remove the normalization factor from the ozone profile data. However, given the emphasis on ozone trends calculations, it would be prudent to demonstrate that these data do not contain a notable step-change in the time series in the Supplemental Information of this manuscript.*

*This paper is extremely well written, and I have only a few mostly minor suggestions, edits, and comments for the authors to address. One note: I couldn't tell if this paper has been submitted to the Tropospheric Ozone Assessment Report 2 Special Issue: https://bg.copernicus.org/articles/special_issue10_1256.html. This paper would be an excellent candidate for the issue given its topic and quality, and it may receive more attention if it is submitted to that Special Issue.*

Response: Thank the reviewer for the positive feedback on the current manuscript. We added a new plot of all ozone profiles used in the analysis (Figure R1) in the revised Supplementary material (L30-32). After removing the normalization factors during the observation period at three Japanese sites, the corrected dataset shows no notable step-change around 2009 in Japanese sites. The related discussion has been added at L122-124 in the revised manuscript.

[Figure]

Figure R1 All O$_3$ profile samples used in the analysis at (a) Hong Kong, (b) Naha, (c) Tsukuba, and (d) Sapporo. Black lines indicate the transition time from CI to ECC ozonesonde at the Japanese stations around 2009.

Thank the reviewer for recommending the Tropospheric Ozone Assessment Report 2 Special Issue. We will apply for it after the ACPD phase.

**Recommendation:** *I recommend this paper's publication once the authors address the mostly minor and technical comments listed below.*

**Specific and Line-by-Line Comments:**

**Line 27** *(Abstract): By "hotspots" do you mean the largest trends? Reconsider the language here.*

Response: Thank you for the suggestion. We intended to discuss the distinct tongue-shaped pattern in top-down direction characterized by high concentrations of O$_3$ extending from the lower stratosphere to the middle troposphere. It has been revised as ozone tongue in the revised manuscript.

**Line 88:** *Because of the time response delay of the ozonesonde sensor, the profiles have at best 100 meter vertical resolution, even if the data are reported every ~5 to 10 meters.*

Response: Thank you for the comment. In the sentence, we have removed the emphasis on the fine vertical resolution of less than 10 m in the revised manuscript. Please see L96-97 in the updated manuscript.

**Line 117**: *I think you mean "underestimation" of the uncertainties rather than "overestimation."*

Response: Thank you for the comment. We agree with the reviewer. The original statement has been rephrased as " However, the transition of the measurement technology from CI to ECC around 2009 led to uncertainties and an overestimation of the long-term $O_3$ trends due to a step-change in the resulting timeseries." in L122-124. Using the corrected ozonesonde data provided by WOUDC during the CI period could lead to overestimating long-term $O_3$ trends (as shown below in Figure R2).

[Figure]

Figure R2. All $O_3$ profile samples were used in the analysis at Naha from (a1) WOUDC and (a2) after removing CFs. Black lines indicate the transition time from the CI to ECC ozonesonde at the Japanese stations around 2009.

*Line 129: This may qualify as more than a minor comment. Both the ascent and descent data are being used here? I do have concerns about using descent profiles given the uncertainty of solution evaporation and loss from the balloon burst/tumbling through the atmosphere.*

Response: Thank you for the comment. We did only use the ascent profiles for the analysis. To clarify, we modify the description as " We limit our analyses of tropospheric and lower-stratospheric $O_3$ profiles to altitudes below 18 km and remove duplicate $O_3$ values during the descent period at the same heights in the time series to prevent redundant measurements as well as the uncertainty of solution evaporation and loss from the $O_3$ sounding balloon burst/tumbling through the atmosphere" in the revised manuscript L140-143.

*Line 216: Change "like" to "such as"*

Response: Thank you for the suggestion. We changed the expressions in L227 in the revised manuscript.

*Line 318: ", while not so clear for the seasonal difference in the middle-upper troposphere." I don't quite follow this part. Please rewrite for more clarity.*

Response: Thank you for the comment. We revised it as "This seasonal difference in the lower troposphere could be attributed to the influence of the East Asia Monsoon as discussed earlier. In the

middle-upper troposphere, there are no such significant seasonal differences among sites.". Please see L332 in the updated version.

**Line 344:** *This is Figure 6b3, not 7c2*

Response: Thank you for the comment. It has been corrected in the revised manuscript.

**Line 345:** *This is Figure 6a2, not 7b1. Please also correct the Figure 6 caption. For example, Hong Kong is (a1-a3), not (a1-c1).*

Response: Thank you for pointing out the error. We have corrected them in the revised manuscript.

**Line 356**: *"Figure .7", remove the period*

Response: Thank you. The period has been removed.

**Line 362:** *Change "than" to "compared to"*

Response: Thank you for the suggestion. We changed it in the revised manuscript.

**Figure 7**: *Not sure if I missed whether all model output for a station is used, or only output coincident with the ozonesonde profiles.*

Response: Thank you for the comment. In this case, all the model output for a station is used. There are two considerations for using all model output rather than the coincident records. First, the monthly output of the model was then used to distinguish the ozone contribution from the troposphere and stratosphere. If the validation of monthly mean output makes sense, it will enhance the robustness of the conclusion we've reached. Second, the weekly launch frequency of the ozonesondes has been validated as reliable in representing long-term $O_3$ trends, as evidenced by comparing them with near-surface $O_3$ trends at hourly time resolution (Liao et al., 2021). Using the monthly mean of weekly resolution ozonesonde observation to validate the monthly mean of 6-hourly time resolution model output on the ozone long-term trend is feasible.

**Figure 7**: *It would be helpful to also include some of the trend values from Table 2 on the figure itself.*

Response: Thanks for the suggestion. We plotted the time series of the monthly mean of the ozonesonde observations and model output of ozone at different levels of the troposphere in Figure 7 to better compare long-term ozone changes with the seasonal variations as well as the absolute values. For a more

accurate comparison of the long-term trends, the seasonal variation should be removed. We decided to discuss the ozone trend in detail in Table 2 by separating the seasons.

*Line 382: I think you mean Figure 4, not Figure 3 here.*

Response: Thank the reviewer for pointing out the error. We have corrected it in the revised manuscript L399.

**Line 521:** *There is an extra "u" in this sentence.*

Response: Thank you. The typo has been removed.

***Data Availability:*** *Are the model output available publicly?*

Response: Thank you for the question. We have uploaded the model output at Zenodo, which can be freely downloaded via https://zenodo.org/records/11093806. The related information has been updated in the revised manuscript in Data Availability Statement (L536-537).

**Reference:**

Liao, Z., Ling, Z., Gao, M., Sun, J., Zhao, W., Ma, P., Quan, J., and Fan, S.: Tropospheric Ozone Variability Over Hong Kong Based on Recent 20 years (2000–2019) Ozonesonde Observation, J. Geophys. Res., 126, e2020JD033054, https://doi.org/10.1029/2020JD033054, 2021.

**Point-by-point responses to the Community Comments#1**

*The impact of stratospheric intrusion on O₃ pollution is an important topic of research, and this paper presents very interesting analysis on this topic. The paper is well organized and for the most part, methodologically sound. More in-depth analysis, as suggested below, is required before this manuscript can be accepted. A minor revision is suggested with a few reasons.*

Response: Thanks for the positive feedback. Please see our point-by-point responses below.

**1. L24, 29, 31:** *ozone has been defined as O₃ on lines 16. Please check similar issues for other abbreviation terms.*

Response: Thank for the suggestion. We have corrected the ozone with abbreviations through the revised manuscript.

**2. L42**: *I disagree with the author's viewpoint that "Stratospheric intrusions and photochemical production are two major contributors to tropospheric ozone". Regional transport can include emissions from urban areas, industrial zones, or other regions that may contain ozone precursors or other compounds that affect ozone formation and degradation. Therefore, considering the impact of regional transport on tropospheric ozone is crucial.*

Response: Thank you for the comment. We agree with the reviewer that regional transport of ozone and ozone precursors is crucial to tropospheric ozone enhancement. The sources for ozone regional transport can be classified into stratosphere and troposphere. The ozone from the stratosphere intrusion and tropospheric photochemical ozone reaction can be further transported from local to other regions. The regional transport of ozone precursors will finally impact the local ozone through photochemical ozone production. Regional transport can be considered as the dynamic driver that contributes based on two major sources, i.e. stratospheric intrusions and photochemical production. To address the discussion of ozone contributors more comprehensively, we mentioned the influence of regional transport in the Part 4 Discussion and Conclusion in the revised manuscript L516-521: "The dynamical and chemical drivers for such long-term tropospheric changes deserve further analysis in the future. Here, we propose several mechanisms based on related research that could potentially contribute to observed tropospheric O₃ enhancements in East Asia. Regional transport is one important contributor to tropospheric O₃ enhancement. Compared with the other two Japanese sites, Naha, to the east of China, is susceptible to regional transport of air pollution from China. The prevailing westerly winds bring O₃-enriched air from eastern China to Naha, resulting in a substantial increase of O₃ from the middle to upper troposphere".

**3. L168-169**: *Is a horizontal resolution of 2.8°× 2.8° too coarse?*

Response: Thank the reviewer for the comment. In the study, we used the global chemistry-climate model EMAC to simulate the 40-year changes of tropospheric ozone. The model considered the interaction of chemistry and dynamic processes between the surface and the middle atmosphere. It is always good to have simulation results with fine time resolution. However, it would be computationally expensive to run such a long-term hindcast simulation with finer spatial resolution on a global range. The current spatial resolution for the simulation is the trade-off between performance and computational capacity. According to the evaluation results in 3.2.1, the EMAC model reasonably simulates the spatial and temporal variations in tropospheric ozone as compared to the ozone observations at the four sounding sites.

**4. L185-186:What is the loss rate of $O_3S$** *tracer? Please provide a calculation method.*

Response: Thank you for the question. The stratospheric ozone tracer $O_3S$ is destroyed in the troposphere as $O_3$ (Roelofs and Lelieveld, 1997; Jöckel et al., 2006). The total chemical loss rate of $O_3S$ in the troposphere is given by:

$$O_3 + OH \rightarrow O_2 + HO_2 \tag{1}$$

$$O_3 + HO_2 \rightarrow 2O_2 + OH \tag{2}$$

$$O_3 + hv \rightarrow O(1D) + O_2 \tag{3}$$

$$O(1D) + H_2O \rightarrow 2OH \tag{4}$$

$$L_{O_{3s}} = k_1[OH] + k_2[HO_2] + \frac{J_3 k_4[H_2O]}{k_2[M] + k_4[H_2O]}$$

In which $k$ and $J$ values represent the reaction rates and photodissociation coefficients, respectively. The square brackets indicate concentrations. The dry deposition of $O_3S$ is calculated the same as $O_3$.

**5. L191-193:** *This looks pretty strange. Considering that the tropopause height decreases with increasing latitude, the author defines 200 hPa as the upper troposphere for Hong Kong and NAHA, while defining 400 hPa as the upper troposphere for Tsukuba and Sapporo. However, the author defines both mid-level and lower-level troposphere as 500 hPa and 850 hPa for all four sites.*

Response: Thank you for the comments. We agree with the reviewer that choosing the same model pressure layers for the middle and low troposphere for four sites at different latitudes can introduce uncertainties. Due to the uneven distribution of pressure layers, it's hard to accurately classify the range of upper, middle, and lower troposphere as our definition for normalized height in observations. Hence, we choose the same specific layer to simply and coherently represent the middle and lower layers in the troposphere for a better comparison with the observed ozone trends. To avoid the uncertainties induced by the layer selection in model results, we add more comparison and analysis based on the time-height cross sections of modeled ozone as well as ozone tracer, as shown in Figures 8, 9, 10, 11.

**6.** *Could you provide the data source of the tropopause height in Figure 2?*

Response: The tropopause height in Figure 2 is calculated by ozonesonde observations based on the World Meteorological Organization lapse rate tropopause definition. The detailed information about calculation is at L148-153 in the manuscript:" Due to the latitudinal differences and the seasonal variations in tropopause height across the four $O_3$-sounding observation sites, it is inappropriate to apply a specific height as the tropopause height. We thus employ the World Meteorological Organization lapse rate tropopause definition to calculate the tropopause height (hereafter called $Z_t$) for each site and $O_3$ profile. The $Z_t$ is defined as the level at which the lapse rate decreases to 2 K km$^{-1}$ or less, provided that the average lapse rate between this level and all higher levels within 2 km does not exceed 2 K km$^{-1}$ (WMO, 1957)." We modified the title of Figure 2 and Figure 4 as "Black dash lines indicate the tropopause heights calculated by observations according to the WMO lapse rate tropopause definition" in the revised manuscript.

*7. Again, ozone has been defined!*

Response: Thank you for the suggestion. We have double-checked and corrected the ozone with $O_3$ in the revised manuscript.

*8. The uncertainty of stratospheric $O_3$ tagging method needs to be discussed.*

Response: Thank the reviewer for the suggestion. The uncertainties of the stratospheric $O_3$ tagging method include: Model Uncertainties: Model uncertainties can arise from inaccuracies in representing atmospheric dynamics, parameterizations of physical and chemical processes, and spatial or temporal resolution; Chemical Kinetics Uncertainties: Chemical reactions involving $O_3$ can have uncertainties in their rate constants and reaction mechanisms. These uncertainties can propagate through the modeling process and affect the accuracy of ozone tagging results; Vertical Transport Uncertainties: The vertical transport of tagged $O_3$ in the atmosphere, particularly between the troposphere and stratosphere, is complex and can be subject to uncertainties in the representation of processes such as convection, mixing, and advection; Sensitivity to Initial Conditions: The results of ozone tagging simulations can be sensitive to the initial conditions used in the atmospheric models. Uncertainties in the initial state of the atmosphere, including ozone concentrations and meteorological parameters, can influence the simulated ozone distribution and its attribution to different sources. As for our EMAC hincast simulation, it considers the interaction of chemistry and dynamic processes between the surface and the middle atmosphere (Jöckel et al., 2016), with the specific dynamics nudging by Newtonian relaxation towards ECMWF ERA-5 reanalysis meteorological data (Hersbach et al., 2020). So, the uncertainties mentioned above such as Chemical Kinetics Uncertainties, Sensitivity to Initial Conditions, and Vertical Transport Uncertainties are maximum optimized.

Since there is no such observation available for the evaluation of the $O_3$ tagging results. We can only infer the performance of tagged $O_3$ by the overall simulated $O_3$. In general, the EMAC model reasonably

simulates the spatial and temporal variations in tropospheric $O_3$ as compared to the $O_3$ observations at the four sounding sites. Consistency between the model and observations suggests that the $O_3$ tagging results remain valuable despite uncertainties related to the model uncertainties. Moreover, the model can effectively be used to investigate the stratospheric ozone contributions to the tropospheric $O_3$ changes.

**Reference:**

Hersbach, H., Bell, B., Berrisford, P., Hirahara, S., Horányi, A., Muñoz-Sabater, J., Nicolas, J., Peubey, C., Radu, R., Schepers, D., Simmons, A., Soci, C., Abdalla, S., Abellan, X., Balsamo, G., Bechtold, P., Biavati, G., Bidlot, J., Bonavita, M., De Chiara, G., Dahlgren, P., Dee, D., Diamantakis, M., Dragani, R., Flemming, J., Forbes, R., Fuentes, M., Geer, A., Haimberger, L., Healy, S., Hogan, R. J., Hólm, E., Janisková, M., Keeley, S., Laloyaux, P., Lopez, P., Lupu, C., Radnoti, G., de Rosnay, P., Rozum, I., Vamborg, F., Villaume, S., and Thépaut, J.-N.: The ERA5 global reanalysis, Q. J. R. Meteorol. Soc., 146, 1999-2049, https://doi.org/10.1002/qj.3803, 2020.

Jöckel, P., Tost, H., Pozzer, A., Brühl, C., Buchholz, J., Ganzeveld, L., Hoor, P., Kerkweg, A., Lawrence, M. G., Sander, R., Steil, B., Stiller, G., Tanarhte, M., Taraborrelli, D., van Aardenne, J., and Lelieveld, J.: The atmospheric chemistry general circulation model ECHAM5/MESSy1: consistent simulation of ozone from the surface to the mesosphere, Atmos. Chem. Phys., 6, 5067-5104, https://doi.org/10.5194/acp-6-5067-2006, 2006.

Jöckel, P., Tost, H., Pozzer, A., Kunze, M., Kirner, O., Brenninkmeijer, C. A. M., Brinkop, S., Cai, D. S., Dyroff, C., Eckstein, J., Frank, F., Garny, H., Gottschaldt, K. D., Graf, P., Grewe, V., Kerkweg, A., Kern, B., Matthes, S., Mertens, M., Meul, S., Neumaier, M., Nützel, M., Oberländer-Hayn, S., Ruhnke, R., Runde, T., Sander, R., Scharffe, D., and Zahn, A.: Earth System Chemistry integrated Modelling (ESCiMo) with the Modular Earth Submodel System (MESSy) version 2.51, Geosci. Model Dev., 9, 1153-1200, https://doi.org/10.5194/gmd-9-1153-2016, 2016.

Roelofs, G.-J., and Lelieveld, J.: Model study of the influence of cross-tropopause O3 transports on tropospheric O3 levels, Tellus. B., 49, 38-55, https://doi.org/10.3402/tellusb.v49i1.15949, 1997.

**Point-by-point responses to the Review Comments #2**

Review of "Causes of growing middle-upper tropospheric ozone over the North West Pacific region" by Xiaodan Ma et al.

**Overall Assessment**

*This study leverages ozonesonde records over a relatively long (>20 year) period for four locations bordering the North-West Pacific Ocean, in comparing a state-of-the-art chemistry climate model (EMAC) in simulating ozone across different latitudes spanning the near-tropics to mid-latitudes, as well as from the surface to the upper-troposphere-lower stratosphere region. The authors confirm good overall agreement in terms of EMAC's ability to capture the seasonality in tropospheric ozone, as well as in many cases relatively large, robust trends/changes that have occurred at each location from the 1990s to 2010s, according to the ozonesonde records. They do this by separating out upper, middle and lower troposphere, abbreviated UT, MT and LT respectively, and computing monthly/seasonal means. By exploiting the stratosphere-tagged ozone tracer ($O_3S$) in the EMAC model, the authors then proceed to quantify the attribution from the stratosphere versus that formed in the troposphere according to the model climatology. A detailed assessment of the stratospheric contribution to recent trends/changes is then performed. Calculation of the residual amount of ozone formed in the troposphere ($O_3T$) helps to then attribute the role of tropospheric production.*

*The work is of clear importance as ozone is a non-well mixed greenhouse gas, with a typical lifetime of ~3 weeks in the free troposphere, and is sensitive to multiple factors, including in situ photochemical production, long-range transport and large-scale dynamics. Uncertainty in what drives/influences tropospheric ozone is therefore large and there is an obvious need to understand this better as there is an obvious radiative forcing and air quality implication to the results, as recognised by the wider literature. Dynamical influence mainly entails changes in the downward transport of ozone from the stratosphere, which is strongly determined by the strength of the Brewer-Dobson Circulation (the residual meridional stratospheric overturning circulation forced by wave-mean flow interactions). Facilitation of such downward transport arises from the action of tropopause folding, which preferentially occurs in association with the subtropical/eddy-driven jets and transient synoptic-scale systems (e.g., cut-off lows). Filamentation of ozone-rich air near the tropopause can subsequently be entrained into the troposphere (so-called 'stratospheric intrusions').*

*This regional-focused study is therefore highly valuable and is complementary to many earlier studies which have a more general focus (e.g., global). The study is logically presented and well executed, fitting well within the scope of Atmospheric, Chemistry and Physics journal. I would recommend publication of the study, following a minor round of revisions as per my general and minor comments which I detail below:*

Response: We would like to thank the reviewer for her/his overall positive assessment of our manuscript and for pointing out the importance of our study for this research field. Thank you also for the valuable comments which have helped improve our manuscript further and which are addressed in detail below.

**General Comments**

*L56: "...Brewer–Dobson circulation promotes stratospheric intrusions..."→I wouldn't say this is true. An enhanced BDC may facilitate an enrichment of ozone into the extratropical lowermost stratosphere that is readily available to be transported down into the troposphere. The development of stratospheric intrusions is instead governed by processes operating on much finer spatial scales, typically synoptic–scale or less, such as associated with the rear-flank of extratropical cyclones/cut-off lows. The authors detail this later (L63–65).*

*The authors I think need to change the wording to reflect that an increase in BDC strength can promote greater seasonal build-up of ozone into the extratropical lowermost stratosphere over winter (Ray et al., 1999; Konopka et al., 2015; Ploeger & Birner, 2016). Subsequent stratospheric intrusions can then facilitate greater stratosphere-troposphere exchange of ozone as a result of this enrichment, particularly around March time when the lowermost stratospheric reservoir of ozone reaches an annual maximum and is seasonally "flushed thereafter" (Hegglin and Shepherd, 2007; Bönisch et al., 2009). This however does depend on changes to each of the deep and shallow branches of the BDC; a strengthening of the deep branch indeed serves to increase lowermost stratospheric ozone. Strengthening of the shallow branch on the other hand favours enhanced transport/mixing of low-ozone air from the tropical upper troposphere (e.g., Fig. 4 in Bönisch et al., 2009).*

*Given the BDC is pertinent to this study, I suggest the authors also introduce this key distinction between the shallow and deep branches and their significance in regard to stratosphere- troposphere exchange of ozone.*

Response: We thank the reviewer for pointing out the inaccurate statement and for the constructive suggestions. We reiterate the original statement as "an enhanced Brewer-Dobson circulation (BDC) can promote greater seasonal build-up of $O_3$ in the extratropical lowermost stratosphere during winter (Ray et al., 1999; Konopka et al., 2015; Ploeger & Birner, 2016). Subsequent stratospheric intrusions can then lead to the increased stratosphere-troposphere exchange of $O_3$ as a result of this enrichment, particularly in spring when the lowermost stratospheric reservoir of $O_3$ reservoir reaches its annual maximum and is seasonally "flushed" thereafter (Hegglin and Shepherd, 2007; Bönisch et al., 2009). However, this

process depends on changes in the BDC's deep and shallow branches. Strengthening of the deep branch increases lowermost stratospheric $O_3$ while strengthening of the shallow branch favors enhanced transport and mixing of low-ozone air from the tropical upper troposphere (Bönisch et al., 2009)". Please refer to L55-64. In addition, a statement of "The shallow branch of BDC is associated with the breaking of synoptic and planetary-scale waves in the subtropical lower stratosphere (Plumb, 2002; Birner and Bönisch, 2011)" was added on L71-72 in the revised manuscript.

*L208-209: "The tropopause folding on the south part of the STJ could lead to more stratospheric intrusion contributions to the ozone tongue."→I think it would help if the authors provided an overlay in Figure 2 of the standard deviation or similar (to highlight interannual variability). Not only because this could provide more useful information, but it would be expected I think that higher standard deviation would support this speculative point. I wonder if the authors could also go a step further and attempt to diagnose the STJ from one year to the next, which would bolster this assertion?*

Response: Thank the reviewer for the valuable comment and suggestion. The standard deviation of wind speed at 200 hPa shows large interannual variability over the mid-latitude around 30ºN, especially in the Northwest Pacific coastal region (Figure R1). Additionally, we have plotted jet frequency and tropopause folding frequency products diagnosed by ETH Zurich to investigate the spatial relation between the subtropical jet stream (STJ) and tropopause folding events (Figure R2). The high occurrence rate of tropopause folding events occurred in the southern part of the subtropical jet frequent region. Schwartz et al. (2014) also highlighted the high occurrence frequencies of double WMO tropopause at North Hemisphere mid-latitude around 30-45ºN are associated with corresponding maxima in jet frequency distributions, contributing to stratosphere-troposphere transport. We have incorporated Figure R2 as Figure S4 in the revised manuscript's supplementary material to support the claim, which we now reformulated in L222-224 to "Tropopause folding events are located preferentially on the southern flank of the STJ, with the associated stratosphere-to-troposphere transport of $O_3$ thus potentially contributing to the observed seasonal lag in the occurrence of the $O_3$ tongues (Figure S4)."

[Figure]

Figure R1. The standard derivation (SD) of wind speed (color shades) retrieved from ERA5 (the fifth generation ECMWF reanalysis) at the 200 hPa level during 1990-2020. Four $O_3$-sounding sites are indicated in the black squares.

[Figure]

Figure R2. Climatological distribution of tropopause folding frequency (shaded color) and jet frequency (contour lines, units: %) during 2000-2018, products provided by ETH Zurich. Four $O_3$-sounding sites are indicated with the red squares.

***Sect 3.2.1 (L337-346):*** *Although EMAC generally overestimates ozone, there is clearly a tendency towards higher overestimation for lower ozone mixing ratios and lower overestimation at higher ozone mixing ratios (which changes in sign for most distributions, such that EMAC typically underestimates*

*ozone at the highest ozone mixing ratios). The authors neglect to mention this so should consider adding description/quantification of this.*

*I also think the authors need to be a little bit careful with the interpretation for instance that EMAC better represents the upper and lower troposphere versus the middle troposphere for Hong Kong/Naha. The RMSE and MAE imply otherwise at face value, though this is because the differences are less systematic and thus compensated for. The total error summed for all soundings versus model values would maybe be larger for the MT but hard to tell as the axes' ranges are smaller for MT and LT compared with UT. Perhaps the authors would like to compute this, add and compare, which would help support their claim?*

Response: Thank you for the comment and suggestion. We have expanded our discussion of ozone underestimation at higher ozone mixing ratios in L365-368 "It is worth noting that although EMAC generally overestimates $O_3$, there is a tendency towards higher overestimation for lower $O_3$ mixing ratios and lower overestimation at higher $O_3$ mixing ratios, especially for the UT $O_3$ at the Tsukuba and Sapporo sites (Figure 6c1, 6d1)." as well as the addition of the discussion for RMSE and MAE using "The root mean standard error (RMSE) and mean absolute error (MAE) of $O_3$ are larger in the UT than in MT and LT." in L358-359.

The total error summed for all soundings can be indicated by the mean error (ME). We addressed the systematic bias by discussing "the majority of data points are located above the 1:1 line at all sites, indicating that the EMAC over-predicts $O_3$ in the troposphere, which agrees with other related studies (Jöckel et al., 2016; Young et al., 2018; Revell et al 2018)." in L356-358. However, since ME reflects only the direction of the bias (over- or underestimation) and can mask large errors if large positive and negative errors cancel out. We complement ME with additional metrics such as $R^2$, RMSE, and MAE to provide a comprehensive assessment of model performance.

**Minor Comments**

**L46:** *"Tropospheric $O_3$ increases of 7% (measured as a partial column between 3-9 km)..."→The authors may wish to specify this increase is of "Free tropospheric $O_3$", thereby excluding potentially different trends in the lower troposphere/ABL.*

Response: Thank you for the suggestion. We have modified it into "Free tropospheric $O_3$" in L46.

**L51-52:** *"..., the counterpart in the middle-upper troposphere..."→"..., the evolution in the middle-upper troposphere..."?*

Response: Thank you for the suggestion. We have changed it into "evolution" in L51.

**L58:** *"a stratospheric chemistry-climate model"→Is this correct or should this be a chemistry-climate model which has a well-resolved stratosphere?*

Response: Thank you for the comment. To clarify what kind of model was used we now write "A study using a coupled atmosphere-ocean model with interactive stratospheric chemistry …" in L64-66. This is to clarify that the chemistry-climate model used had no representation of tropospheric chemistry, a key distinction to a chemistry-climate model with a well resolved stratosphere.

**L118-120:** *"The research from the cross-evaluation of OMI data and the ozonesonde observation in Japan sites shows that CI ozonesonde measurements are negatively biased relative to ECC measurements by 2–4 DU compared with the OMI data (Bak et al., 2019)".→Unclear if the bias of 2-4 DU is for total column, or partial column only in troposphere.*

Response: Thank you for the comment. We clarified it as "The research from the cross-evaluation of OMI data and the ozonesonde observation in Japan sites shows that CI ozonesonde measurements of tropospheric ozone columns are negatively biased relative to ECC measurements by 2–4 DU compared with the OMI data" in L124-127.

**L199:** *"Figure 2 depicts the climatologically vertically resolved tropospheric $O_3$ distribution with respect to months"→Suggested sentence revision: "Figure 2 depicts the monthly climatological vertically resolved tropospheric $O_3$ distribution throughout the year".*

Response: Thank you for the suggestion. We rewrite it as "Figure 2 depicts the monthly climatological vertically resolved tropospheric $O_3$ distribution throughout the year" in L213.

**Figure 2:** *Yellow colour shading missing for Tsukuba. This needs checking.* **Figure S1:** *Same as issue as above for panels c1 (Tsukuba) and d1-d2 (Sapporo).*

Response: Thank you for the comment. We double-checked the code and plot. The missing yellow colour shading is due to discontinuous changes in $O_3$ concentrations in MT and UT.

**L208:** *"Figure S2"→Change to Figure S1 and reverse order of existing Figures S1 and S2 so that they are in order.*

Response: Thank you for the suggestion. The order has been corrected and Figure S2 in L208 now changed into Figure S3 due to the additional two plots added.

***Figure 3 Caption:*** *I think it would be helpful to remind the reader of what is used to represent upper, middle and lower troposphere. I'm not entirely convinced that each should be best represented by a single pressure level though, particularly for the lower troposphere where ABL and free tropospheric ozone amounts and variability (including its trends) can differ substantially. Also, are the time periods the same as in Figure 2?*

Response: Thank you for the comment. As for observations, we normalize the height of each $O_3$ profile into 0~1 by dividing the altitude by the tropopause height Zt to better compare $O_3$ levels and trends at different latitudes within the troposphere. The upper troposphere (UT) is then defined by the normalized height (Z/Zt) range between 0.7 and 0.9. The middle troposphere (MT) and lower troposphere (LT) are 0.4~0.6 and 0~0.2 Z/Zt, respectively.

The periods of observation data in Figure 3 are the same as in Figure 2. The detailed information about periods is indicated in Table 1. The ozonesonde observation data for Naha and Sapporo are available from 1990 to 2017, and from 1990 to 2020 for the Tsukuba site. We corrected the Figure 2 Caption as "Figure 2. Month-height cross sections of monthly mean $O_3$ at four $O_3$-sounding sites, (a) Hong Kong, (b) Naha, (c) Tsukuba, and (d) Sapporo, from 1990 to 2017/2020 (2000 to 2020 for Hong Kong). Black dash lines indicate the multi-year average tropopause height calculated by observations according to the WMO lapse rate tropopause definition."

The Figure 3 Caption has been rewritten as "Figure 3. Long-term changes of $O_3$ in the Upper Troposphere (defined as 0.7-0.9 tropopause normalized height, first column), Middle Troposphere (defined as 0.4-0.6 tropopause normalized height, second column), and Lower Troposphere (defined as 0-0.2 tropopause normalized height, third column) in boreal spring (MAM, red lines), summer (JJA, yellow lines), autumn (SON, black lines), and winter (DJF, blue lines) at Hong Kong (a1-a3), Naha (b1-b3), Tsukuba (c1-c3), and Sapporo (d1-d3). Trends with a star symbol (*) indicate significance at the 95% confidence level."

***Figure 4c1 and 4d1-2:*** *Same issue as before with contour shading, certain colours missing.*

Response: Thank you for the comment. The missing of colour shading is due to discontinuous changes in $O_3$ concentrations in the MT and UT.

***Figure 4 Caption:*** *"Dashed lines in the i–l represent…"→Typo? I assume this refers to the c panels for each site?*

Response: Thank you for the comment. It has been corrected into "a3-d3" in the revised manuscript L301.

***L280:*** *"Figure S1"→See L208 comment.*

Response: Thank you for the comment. It has been corrected as "Figure S4" in our revised manuscript L224.

*L287: "20 to 40 ppb."→Difficult to tell as colour scale saturates above 20 ppb? Should this be extended?*

Response: Thank you for the suggestion. We agree that the current color scales in Figure 4a3-d3 do not adequately show the absolute changes in $O_3$ values. Therefore, we have removed the conclusion based on this figure. The details of the $O_3$ changes can be found in Table 3.

*L288: "...Hong Kong shows more significant $O_3$ changes in the lower troposphere"→This seems true in the ABL (~<1.6 km) and above what is likely 850 hPa (which I think the authors still use to represent the lower troposphere) for April and August (~2-4 km). However, there is a region in 3 between which is statistically insignificant throughout the calendar year. I don't feel the text gives full justice to the nature of the results shown in Figure 4a3.*

Response: Thank you for the comment. To better compare $O_3$ levels and trends at different latitudes within the troposphere, we normalize the height of each $O_3$ profile into 0~1 by dividing the altitude by the tropopause height (Zt) (P4L169-172). The lower troposphere (LT) is defined as 0~0.2 Z/Zt. For the Hong Kong site, the tropopause height is around 17km throughout the year. The LT for Hong Kong is between 0-3.4 km in this study's definition.

*L333: "3.2 Quantification of stratospheric intrusion versus tropospheric production using EMAC simulations"→It seems only later subsection 3.2.3 directly relates to this heading (the later sub-heading is essentially worded the same).*

*May I suggest to the authors re-writing the section 3.2 header to encompass the model- measurement evaluation of tropospheric ozone first in 3.2.1, something along the lines of: "3.2 Comparison with observations and stratospheric versus tropospheric attribution using EMAC simulations"*

Response: Thank the reviewer for the suggestion. We now use the new header "3.2 Comparison with observations and stratospheric versus tropospheric attribution using EMAC simulations" to better encompass the contents. Please see L349.

*L344: "Figure 7c2"→I think this refers to Figure 6c2. L345: "Figure 7 b1"→I think this refers to Figure 6b1. L356: "Figure .7"→Figure 7*

Response: Thank the reviewer for pointing out the error. They are all corrected in the L362-363, and 368.

*L375-376:* *"Bold indicates the agreement with the observations for significance and the sign of the trend"→Not all bold entries have asterisks next to both the model and observed trend. Is there a mistake or did I misinterpret the description?*

Response: Thank you for the comment. The trend with the same sign and both not significant are also indicated by bold (L394-395).

*L338:* *"The EMAC simulations of O₃ at different portions of the troposphere..."→Revised sentence suggestion: "The EMAC simulations of O₃ for different altitude ranges in the troposphere..."*

Response: Thank the reviewer for the suggestion. It has been corrected in L355.

*L389:* *"Dots in the i-l represent..."→Again, not clear what i-l stands for. Is it a typo?*

Response: Thank you for the suggestion. The "i-l" for clarification has been changed to "a3-d3" in L407.

*Figure 8d1-d2→Check colour scale. Seems that yellow filled contour is missing in Figure 8d1 and possibly cyan filled contour missing in both.*

Response: Thank you for the suggestion. We *double*-checked the code and plot. The missing of the colour shading is because of the discontinuous changes in MT and UT O₃ values.

*L406-407:* *"Furthermore, together with dynamical processes such as tropopause folding in the vicinity of the subtropical jet (Baray et al., 2000), stratospheric O₃ is transported downward into the troposphere."→The authors should also add mention of the seasonal lifting of the which will naturally contribute to entrainment of ozone-rich air from the stratosphere into the troposphere (e.g., Monks, 2000).*

Response: Thank the reviewer for the suggestion. The discussion about the seasonal lifting of tropopause has been added in L424-425 "The seasonal lifting of the tropopause will naturally contribute to the entrainment of O₃-rich air from the stratosphere into the troposphere (Monks, 2000)".

*Figure 9d2→Yellow colour missing.*

Response: Thank you for the comment. The missing of the yellow color is because of the discontinuous changes in UT O₃ values.

*L434:* *"3.2.3 Quantification of stratospheric intrusion vs. tropospheric production using EMAC"→This is almost word for word identical to the 3.2 heading on L333. It is more fitting here,*

*but I would simplify by heading it as follows: "3.2.3 Attribution of EMAC tropospheric $O_3$ changes: $O_3S$ vs. $O_3T$"*

Response: Thank the reviewer for the suggestion. We changed the subtitle as suggested in L453.

**L442-443:** *"The contribution of $O_3S$ to observed $O_3$ increases by up to 96% and 40% in the middle-upper troposphere during winter and summer."→I'm a little unclear where these numbers are from. Are they directly shown in Table 3? 96 % is shown for Sapporo in JJA and 40 % for Tsukuba in 4 JJA (UT), could that be what the authors refer to? It needs to be clearer exactly what season, site and region of troposphere this is (the authors state middle-upper troposphere so reading this implies the quantified contribution is a conflation of the two regions).*

Response: Thank you for the comments. We now clarified it as "The contribution of $O_3S$ to observed $O_3$ increases by up to 96% at Sapporo during DJF and 40% at Tsukuba during JJA in the upper troposphere (Table 3)" (L461-462).

**Table 3:** *It seems to me there is a missing set of numbers and that is the total change in $O_3$ between the 2010s and 1990s for the EMAC model (noting that similar such values in Table 2 are linear trend changes and not the difference between the later and earlier period). These numbers would be good to have so the reader can fully appreciate attribution from the stratosphere versus troposphere without having to do the math.*

*There appears to be a few inconsistencies in the Table. Presumably the absolute and percentage change values should be equivalent in sign for each season, location and each of UT, MT and LT, which is not always the case. This needs checking.*

Response: Thank you for the suggestion. We have double-checked Table 3 and added the tropospheric $O_3$ changes as shown in the following table. Please note that the percentage change values do not always have the same sign as the absolute change. This is seen in Hong Kong and Tsukuba for DJF in the UT, where the total ozone changes are negative, the $O_3S$ and $O_3T$ changes though are negative (negative divided by negative yields positive) and positive (negative divided by positive yields negative), respectively.

Table 3. Tropospheric $O_3$ changes and contributions from $O_3S$ and $O_3T$ to changes of tropospheric $O_3$ between the 2010s and 1990s at the upper, middle, and lower troposphere (UT, MT, and LT) in different seasons. The percentage contributions of $O_3S$ and $O_3T$ to $O_3$ changes are listed in the parentheses.

| Station | | O₃ changes (ppb) | | | | O₃S changes (ppb) | | | | O₃T changes (ppb) | | | |
|---|---|---|---|---|---|---|---|---|---|---|---|---|---|
| | | MAM | JJA | SON | DJF | MAM | JJA | SON | DJF | MAM | JJA | SON | DJF |
| Hong Kong | UT | 3.55 | 12.53 | 7.09 | -0.40 | −2.03 (−57%) | 1.44 (11%) | 1.41 (20%) | −3.44 (860%) | 5.58 (157%) | 11.09 (89%) | 5.69 (80%) | 3.04 (−760%) |
| | MT | 6.35 | 9.22 | 7.50 | 0.32 | 1.30 (20%) | 0.96 (10%) | 1.23 (16%) | −2.84 (−888%) | 5.06 (80%) | 8.27 (90%) | 6.27 (84%) | 3.16 (988%) |
| | LT | 9.62 | 11.47 | 6.28 | 2.10 | 0.88 (9%) | 0.10 (1%) | −0.13 (−2%) | 1.24 (59%) | 8.73 (91%) | 11.37 (99%) | 6.41 (102%) | 0.86 (41%) |
| Naha | UT | 5.94 | 14.76 | 7.76 | 1.31 | 1.05 (18%) | 3.81(26%) | 2.98 (38%) | −1.87 (−143%) | 4.90 (82%) | 10.95 (74%) | 4.78 (62%) | 3.18 (243%) |
| | MT | 8.52 | 6.29 | 6.74 | 2.19 | 2.32 (27%) | 0.08 (1%) | 1.10 (16%) | −1.03 (−47%) | 6.19 (73%) | 6.22 (99%) | 5.64 (84%) | 3.22 (147%) |
| | LT | 5.86 | 3.32 | 1.75 | 1.71 | 2.35 (40%) | −0.19 (−6%) | 0.07 (4%) | 0.73 (43%) | 3.51 (60%) | 3.51 (106%) | 1.68 (96%) | 0.98 (57%) |
| Tsukuba | UT | 10.65 | 11.45 | 6.35 | -2.08 | 7.33 (69%) | 4.23 (40%) | 2.19 (34%) | −4.59 (221%) | 3.32 (31%) | 7.22 (60%) | 4.15 (66%) | 2.51 (−121%) |
| | MT | 4.54 | 7.39 | 5.18 | 2.74 | 1.50 (33%) | 2.10 (28%) | 1.39 (27%) | 0.51 (19%) | 3.04 (67%) | 5.29 (72%) | 3.79 (73%) | 2.23 (81%) |
| | LT | 2.50 | 2.17 | 0.24 | 0.98 | 1.27 (51%) | 0.44 (20%) | 0.94 (392%) | 0.90 (92%) | 1.22 (49%) | 1.74 (80%) | −0.70 (−292%) | 0.08 (8%) |
| Sapporo | UT | 8.66 | 8.58 | 5.11 | 4.82 | 6.85 (79%) | 3.19 (37%) | 2.00 (39%) | 4.65 (96%) | 1.82 (21%) | 5.40 (63%) | 3.11 (61%) | 0.17 (4%) |
| | MT | 3.80 | 5.73 | 3.88 | 2.27 | 1.60 (42%) | 1.59 (28%) | 1.31 (34%) | 1.62 (71%) | 2.20 (58%) | 4.14 (72%) | 2.57 (66%) | 0.65 (29%) |
| | LT | 2.37 | 2.80 | 0.27 | 0.60 | 1.19 (50%) | 0.35 (13%) | 0.71 (263%) | 0.69 (115%) | 1.18 (50%) | 2.45 (87%) | −0.45 (−163%) | −0.09 (−15%) |

*L469:* *"110°N to150°N"→"110°E to 150°E"*

Response: Thank you for pointing out the error. It now has been corrected in L481.

*Figure 11:* *I wonder if the authors could add an equivalent Figure but expressed as a percentage (this could go in the supplementary information at the end). This would be extremely handy to compare and should be minimal effort for the authors to produce and include.*

Response: Thank you for the suggestion. A plot of latitude-pressure cross sections of difference of $O_3S$, and $O_3T$ relative to climatological $O_3$ (%) between the 2010s and 1990s (Figure R3) has been added as Figure S6 in supplementary.

[Figure]

Figure R3. Latitude-pressure cross sections of difference of $O_3S$, and $O_3T$ relative to climatological $O_3$ (%) between the 2010s and 1990s along the Northwest Pacific region (zonal mean over 110°E to 150°E) in four seasons. Black lines indicate the climatological distribution of $O_3S$, and $O_3T$, respectively. Red solid lines denote the tropopause height. Dots represent the layer with statistically significant changes according to a paired two-sided t-test ($p < 0.05$).

*L488-490:* *"On the other hand, ozone originating from the stratosphere dominates the large portion of middle-upper tropospheric $O_3$ enhancement by up to 96% and 40% in the mid-latitude during winter and spring.→Again, these numbers are very specific and I'm not sure if they are plucked directly from Table 3 or not. If the latter, it is surely better to give a*

*range that encompasses both MT and UT, including both Tsukuba and Sapporo as I think the authors are using to represent mid- latitudes.*

Response: Thank you for the suggestion. We modified the conclusion as "On the other hand, $O_3$ originating from the stratosphere dominates the large portion of middle-upper tropospheric $O_3$ enhancement by 19-96% and 28-40% in the mid-latitude during winter and spring "(L500-501).

**L518:** *"With increasing greenhouse gasses..."→"With increasing greenhouse gases..."*

Response: Thank you for the comment. It has been corrected.

**L521:** *"...facilitating downward transport to the troposphere u under the influence of the Pacific jet..."→"...facilitating downward transport to the troposphere under the influence of the Pacific jet..."*

Response: Thank you for the comment. The typo has been corrected.

**References**

Birner, T., and Bönisch, H.: Residual circulation trajectories and transit times into the extratropical lowermost stratosphere, Atmos. Chem. Phys., 11, 817-827, https://doi.org/10.5194/acp-11-817-2011, 2011.

Bönisch, H., Engel, A., Curtius, J., Birner, T., and Hoor, P.: Quantifying transport into the lowermost stratosphere using simultaneous in-situ measurements of SF6 and CO2, Atmos. Chem. Phys., 9, 5905-5919, https://doi.org/10.5194/acp-9-5905-2009, 2009.

Hegglin, M. I., and Shepherd, T. G.: O3-N2O correlations from the Atmospheric Chemistry Experiment: Revisiting a diagnostic of transport and chemistry in the stratosphere, J. Geophys. Res. Atmos., 112, https://doi.org/10.1029/2006JD008281, 2007.

Konopka, P., Ploeger, F., Tao, M., Birner, T., and Riese, M.: Hemispheric asymmetries and seasonality of mean age of air in the lower stratosphere: Deep versus shallow branch of the Brewer-Dobson circulation, J. Geophys. Res. Atmos., 120, 2053-2066, https://doi.org/10.1002/2014JD022429, 2015.

Monks, P. S.: A review of the observations and origins of the spring ozone maximum, Atmos. Environ., 34, 3545-3561, https://doi.org/10.1016/S1352-2310(00)00129-1, 2000.

Ploeger, F., and Birner, T.: Seasonal and inter-annual variability of lower stratospheric age of air spectra, Atmos. Chem. Phys., 16, 10195-10213, https://doi.org/10.5194/acp-16-10195-2016, 2016.

Plumb, R. A.: Stratospheric Transport, J. Meteorol. Soc. Jpn., 80, 793-809, https://doi.org/10.2151/jmsj.80.793, 2002.

Ray, E. A., Moore, F. L., Elkins, J. W., Dutton, G. S., Fahey, D. W., Vömel, H., Oltmans, S. J., and Rosenlof, K. H.: Transport into the northern hemisphere lowermost stratosphere revealed by in situ tracer measurements, J. Geophys. Res. Atmos., 104, 26565-26580, https://doi.org/10.1029/1999JD900323, 1999.

Schwartz, M. J., Manney, G. L., Hegglin, M. I., Livesey, N. J., Santee, M. L., and Daffer, W. H.: Climatology and variability of trace gases in extratropical double-tropopause regions from MLS, HIRDLS, and ACE-FTS measurements, J. Geophys. Res. Atmos., 120, 843-867, https://doi.org/10.1002/2014JD021964, 2015.